



# 1 Global basic landform units derived from multi-source digital elevation

# 2 models at 1 arc-second resolution

Xin Yang[1,2,4†], , Sijin Li[1,2,4†], Junfei Ma[1,2,4], Yang Chen[1,2,4], Xingyu Zhou[1,2,4], Fayuan Li[1,2,4], Liyang Xiong[1,2,4], Chenghu
Zhou[3], Guoan Tang[1,2,4*] & Michael E. Meadows[5,6*]
† These authors contributed equally to this work.
* Co-corresponding authors: Guoan Tang tangguoan@njnu.edu.cn; Michael E Meadows michael.meadows@uct.ac.za
[1]School of Geography, Nanjing Normal University, Nanjing, 210023, China
[2]Key Laboratory of Virtual Geographic Environment (Nanjing Normal University), Ministry of Education, Nanjing,
210023, China
[3]Institute of Geographical Information Science and Natural Resources, Chinese Academy of Science, Beijing, 100101,
China
[4]Jiangsu Centre for Collaborative Innovation in Geographical Information Resource Development and Application, Nanjing
210023, China
[5]School of Geography and Ocean Sciences, Nanjing University, Nanjing 210023, China
[6]Department of Environmental & Geographical Science, University of Cape Town, Rondebosch 7701, South Africa

**Abstract.** Landforms are fundamental components of the Earth surface, providing the base on which surface processes operate. Understanding and classifying global landforms, which record the internal and external dynamics of the planet's evolution, constitutes a critical aspect of Earth system science. Advances in Earth observation technologies have enabled access to higher resolution data, for example remote sensing imagery and digital elevation models (DEMs). However, landform data with a resolution of approximately 1 arc-second (approximately 30 m) are lacking at the global scale, which limits the progress of geomorphologic studies at finer scales. Here, we propose a novel framework for global landform classification and release a unique dataset called Global Basic Landform Units (GBLU), which incorporates a comprehensive set of objects that constitute the range of landforms on Earth. Constructed from multiple 1 arc-second DEMs, GBLU ranks among the highest-resolution global geomorphology datasets to date. Its development integrates geomorphological ontologies and key derivatives to strike a balance between mitigating local noise and preserving valuable landform details. GBLU categorizes the Earth's landforms into three levels with 26 classes, yielding discrete vector units that record landform type and distribution. Comparative analyses with previous datasets reveal that GBLU enhances capture of landform details, enabling more precise depiction of geomorphological boundaries. This refinement facilitates the identification of novel spatial disparities in landform patterns, exemplified by marked contrasts between Asia and other continents, and highlights the distinct prominence of China in terms of landform diversity. Given that the fundamental data resolution of GBLU accords well with available remote sensing datasets, it is readily incorporated into analytical workflows, exploring the relationship between landforms, climate and land cover. The full data set is available on the Deep-time Digital Earth Geomorphology platform and Zenodo (Yang et al., 2024; https://doi.org/10.5281/zenodo.13187969).

## 1. Introduction

Approaches to geomorphology vary, and include research on, for example, processes, materials, hazard and risk, and chronology, but the essential basis of all of these studies is the *landform* (Evans, 2012), which can be regarded as the 'final surface status' resulting from the combined influence of various forces. The morphology of landforms and their associated evolutionary processes have long been a source of fascination, leading ultimately to the development of the formal science of geomorphology (MacMillan and Shary, 2009). Classifying and mapping the Earth's surface into landform types according to morphological characteristics is a primary means of understanding surface patterns and processes on planet Earth (Evans, 2012; Xiong et al., 2022) and advancement in this field has potential benefits for the more efficient allocation of global resources to promote sustainable development (Dramis, 2009).

Traditional landform mapping primarily relies upon manual interpretation based on the topographic maps and aerial photographs supported by field investigations (Drăguţ and Blaschke, 2006; Hammond, 1954; Iwahashi et al., 2018; Pennock et al., 1987). However, a series of technological developments has facilitated the automation of landform classification in recent decades, largely dependent on topographic derivatives calculated from DEMs, such as slope, aspect, relief, curvature, roughness (Amatulli et al., 2018; Dyba and Jasiewicz, 2022; Jasiewicz and Stepinski, 2013). With the development of earth observation systems and



DEM refinement, several global landform datasets based on this framework have been proposed using various data sources and at
different levels of spatial resolution (Florinsky, 2017; Iwahashi and Yamazaki, 2022). Using a decision tree algorithm and 1-km
SRTM30 data, Iwahashi and Pike (2007) generated a global terrain classification gridded dataset containing 16 undefined
topographic types determined by slope gradient, local convexity, and surface texture. Relying on elevation and the standard deviation
of elevation, Drăguţ and Eisank (2012) adopted an object-based method to automatically classify global landforms from SRTM data
resampled to 1 km. Meanwhile, Iwahashi et al. (2018) improved their previous work and established 15 landform classes based on
MERIT DEM.  To further eliminate issues involved in detecting narrow valley bottom plains, metropolitan areas, and slight inclines
in otherwise largely flat plains, Iwahashi and Yamazaki (2022) introduced the elevation above the nearest drainage line measure,
and achieved landform classification based on a DEM at 90m resolution. However, as the authors stated, unsupervised classification-
based methods to perform higher-resolution global landform classification require an international team with knowledge of
geomorphological development in a variety of climatic and physiographic settings. In addition, at regional and/or global scales,
several researchers have achieved automated landform classification following the Hammond procedure (Gallant et al., 2005;
Karagulle et al., 2017; Martins et al., 2016). All these datasets have provided valuable resources to explore surface patterns, and
also played important roles in supporting related disciplines such as hydrology, pedology, and ecology among others.
However, shortfalls remain in current landform classification research and require attention to the  following points. Firstly,
previous studies have adopted relatively coarse resolution DEMs, resulting in an inaccurate depiction of topographic information.
Recent developments in Earth observation technology have concentrated on the deployment of digital elevation models (DEMs),
which contain abundant geometric information about surface relief (Drăguţ and Eisank, 2011), although the approach and methods
of implementing landform classification have not kept pace with advances in DEM resolution and quality. Nevertheless, higher
DEM data resolution can be regarded as a double-edged sword, in that it at once provides the opportunity for landform mapping at
a finer scale while at the same time increasing the challenge of reducing the noise effect (Jasiewicz and Stepinski, 2013) and
maintaining the integrity of the identified landforms. Secondly, at the global scale, diverse and complex environmental factors have
shaped different types of landforms that pose substantial challenges to the generalizability of classification methods (Li et al., 2020).
With increasing human impact on landforms, a re-evaluation of landform classification that takes advantage of an increasingly
potent digital database and ongoing improvements in human understanding of landform evolution and processes seems opportune.
Finally, landform information obtained from a particular metric is derived at a particular spatial scale, determined jointly by the
DEM resolution and window size in the neighborhood analysis,  giving rise to uncertainties in the landform classification results.
Therefore, the development of innovative classification approaches and systems based on high resolution DEMs remains a
priority for research on global landforms. In this study, we conduct a classification and mapping of global landforms based on a
DEM at 1 arc-second resolution. We focus on the classification of basic landforms that emphasizes morphological differences and,
in so doing, we present the practical expression of landform ontology at the global scale that offers valuable insights into the Earth's
surface structure comprising the constellation of landform types and their boundaries. The objectives of this research are: (1) to





construct a global classification system for landforms that integrates geomorphological knowledge, (2) to design a novel framework
for global basic landform classification, (3) to develop an automated classification and mapping model for global landforms, and
(4) to make available a comprehensive high-resolutiojn dataset of global landform units.
**2. Methodology**
**2.1 Hierarchical classification system and data**
In aiming to provide a comprehensive classification of landforms at the global scale, our study encompasses all terrestrial
regions worldwide, including islands and polar areas. Identifying landform objects and constructing a classification system is a
preliminary and significant step in geomorphological and landform classification studies. It is crucial to recognize that landforms
not only represent assemblages of quantitative characteristics but also convey the basic human understanding of nature (Smith and
Mark, 2001). For example, the identification of what is acknowledged as a 'mountain' is as much a product of human perception as
of its natural characteristics (Smith and Mark, 2003), thus emphasizing the importance of incorporating human understanding into
landform classification and mapping. Therefore, we focus here on the classification of basic landforms that emphasizes
morphological differences that are not only perceptible to humans but also constitute vital components in the analysis of surface
environments. In taking into consideration the complexity of global landform characteristics, the classification criteria should satisfy
the following requirements: (1) the determined classes should be globally applicable; (2) the setting of the landform types should
conform with the current knowledge domain of geomorphology; and (3) specific criteria should be able to be interpreted and applied.
In employing existing landform classification principles (Zhou et al., 2009), here we propose the set of criteria for basic landform
classification. The new criteria integrate the typical rules of landform classification with indices proposed in this work, and are
aimed at reflecting human knowledge in a quantitative way. We establish a hierarchical classification system comprising 3 levels
and 23 classes (Table A1), thereby advancing a structured framework for understanding Earth's diverse landscapes. The first-level
(L1) types are defined as 'plain' and 'mountain', reflecting the most fundamental morphological characteristics of landforms. Plains
and mountains are the most direct reflection of the combined effects of geomorphological processes and profoundly influence
biological activities. This classification perspective aids researchers in conducting macro-scale studies. At the second level (L2),
plain landforms retain their labels to guarantee completeness of the classification system, and are further divided into low-altitude,
middle-altitude, high-altitude, and highest-altitude plains based on elevation. Mountains are subdivided at L2 into hills and other
mountains with varying degrees of relief. At L3, we provide a further detailed classification of hills and mountains based on elevation.
To attain global coverage, we utilize three DEM datasets (Table 1). These datasets are publicly available for access and have
been widely used in geomorphological studies, ensuring their accuracy and validity. In this work, the 'Forest and Buildings removed
Copernicus DEM' (FABDEM) (Hawker et al., 2022) is the primary data for latitudes 60°S-80°N. This dataset is the first bare-earth
DEM dataset at a global scale at 1 arc-second (approximately 30-meter) resolution, developed using machine learning techniques
from Copernicus DEM. By eliminating the bias resulting from building and vegetation heights, some terrain features, such as slope,



aspect, and watersheds, can be estimated more accurately, which is of significant benefit in landform classification. Meanwhile, the
Advanced Land Observing Satellite (ALOS) World 3D - 30 m (AW3D30) (Tadono et al., 2014) dataset is used to supply data for
the area the missing from FABDEM. In addition, to avoid the negative impact of ocean pixels on landform classification results, the
OpenStreetMap (OSM) Land Polygon was utilized as a mask to eliminate the sea.
**Table 1. Data sources and attributes**

|  | FABDEM | AW3D30 V3.2 | REMA |
|---|---|---|---|
| Spatial Coverage | 60°S-80°N | 82°S-82°N | 56°S-88°S |
| Spatial Resolution | 1 arc-second | 1 arc-second | 32 m |
| Vertical Accuracy | <4 m | 4.4 m (RMSE) | 4 m (RMSE) |
| Release Date | 2021 | 2021 | 2022 |
| Data link | https://data.bris.ac.uk/data/datas et/s5hqmjcdj8yo2ibzi9b4ew3sn | https://www.eorc.jaxa.jp/ALOS/jp/da taset/aw3d30/aw3d30_j.htm | https://www.pgc.umn.edu/ data/rema/ |

**2.2 Global landform classification method**
In this study, we propose a knowledge-guided framework and provide the corresponding implementation workflow. The
proposed method of global landform classification has a hierarchical structure, involving data pre-processing, identification of
mountains and plains, calculation of the mountain uplift index (SUI), landform classification, and post-processing. Figure 1
illustrates the workflow. The following sections provide details that should allow users to reproduce our results. In this study, we
built factor calculation and landform classification models based on tools in ArcGIS Pro. A detailed description of the step-by-step
procedures follows below.

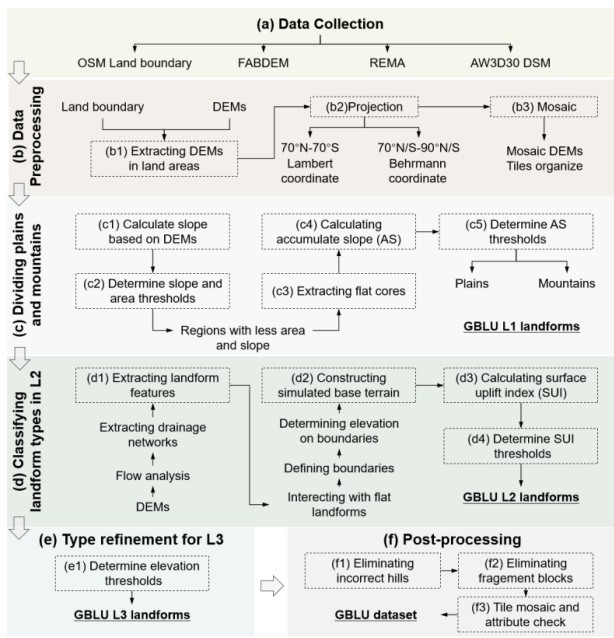


**Figure 1. Workflow for global landform classification used in this study.**



**2.2.1 Data pre-processing**

As shown in Figure 1b, data pre-processing focuses primarily on land area extraction and data merging. We use the OSM land polygon as the land mask to eliminate the marine pixels that negatively influence subsequent processes. To improve processing efficiency, the original DEM elements with size of 1×1 degree are mosaiced to tiles of 10×10 degrees. Meanwhile, due to the requirement of calculating landform derivatives, we determine the projection principles as follows: data from latitudes below 70° are transposed onto the Behrmann projection, and the remaining data are transported onto the Lambert azimuth equal-area projection.

**2.2.2 Identifying plains and mountains**

Landforms represent the most fundamental elements of the Earth's terrestrial surface and reflect both internal and external forces acting over time. Identifying and distinguishing contrasting plains and mountains represents the initial step in basic landform classification and mapping. We have designed a practical framework based on landform ontology to classify plains and mountains. The plains can be separated into core, transition and boundary, whereby the core represents areas with the most typical flat characteristics, i.e. very low relief. Transitions have plain cores but also contain sloping elements, i.e. areas that in part satisfy their classification as plain but also exhibit sloping characteristics not typical of plain. Misclassifications usually occur in transition areas due to their atypical characteristics. Meanwhile, the boundary represents the part of the plain area where the geomorphological semantics and labels change to the mountain.

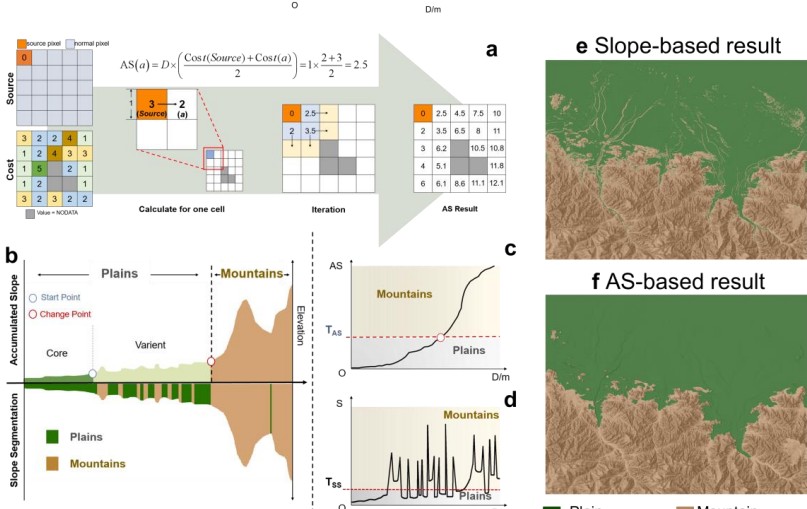

**Figure 2. Illustration of calculation methods. a** schematic diagram of the cost-distance algorithm. **b** profile reflecting landform composition according to the proposed conceptual model of plains, segmented based on the slope; **c** calculated result of the AS and **d** calculated result of slope, where $T_{AS}$ is the threshold of AS, and $T_{SS}$ is the threshold of surface slope. For Figures 2c and d, areas smaller than the threshold are classified as plains (marked in green), while the remaining areas are classified as mountains (marked in brown). **e and f** comparison of the AS and slope indicators in the division of plains and mountains.

Firstly, we regard the areas with low slope angles as the plain cores. Here, the slope threshold ($T_1$) is recommended to be set as 1.5-3 degrees according to our global pre-assessment experiments. Areas where the slope angle lies below the threshold $T_1$ are classified as plain cores. Secondly, we employ the accumulated slope (AS) as representing the different slope attributes of





landforms. The AS is calculated as the minimum cumulative cost of each position to the nearest landform core along a specific path
(Sechu et al., 2021). The algorithm follows the geospatial analysis principle whereby the lowest cost is computed through the
creation of least-cost paths between cores and general positions. The tool of distance accumulation in ArcGIS Pro can achieve this
calculation. In its implementation, this algorithm employs an iteration starting from the cell closest to the cores and follows the
calculation principle shown in Figure 2a to compute the minimum cumulative cost of each cell to the core. The completed area is
then expanded until all grids are associated with increasing costs. Segmenting landforms through the determination of the thresholds
for landform derivatives is one of the most common methods used in geomorphological studies and achieves the most direct
integration of geomorphological knowledge and expertise. As shown in Figure 2b, due to differences in topographic characteristics
between plains and mountains, the AS has a low rate of increase in the areas classified as plains and a high rate of increase in rugged
areas. This phenomenon reduces the difficulty of determining an appropriate AS threshold, which can be achieved by searching for
abrupt changes in the AS profile. In this step, taking into consideration the geomorphological perspective, the threshold of AS ($T_2$)
is recommended to be 1500-2000 based on the pre-experimental results conducted on numerous samples worldwide. This threshold
range is provided as a reference, but needs to be determined by integration with expert knowledge within different geomorphic
regions. In some cases, it may exceed the recommended threshold range. Areas where the AS value is less than $T_2$ are regarded as
plains, and the remaining areas are mountains. Through the above segmentation, we can obtain the boundary of plain and construct
the complete plain area consisting of core, variant and boundary. As shown in Figures 2e and f, this novel workflow exaggerates the
difference between the plains and mountains and converts the local slope into an indicator of global landform characteristics. This
novel method avoids the negative effect of local window analysis and is beneficial for maintaining the landform semantics for each
block.
**2.2.3 Classifying landform types in level 2**

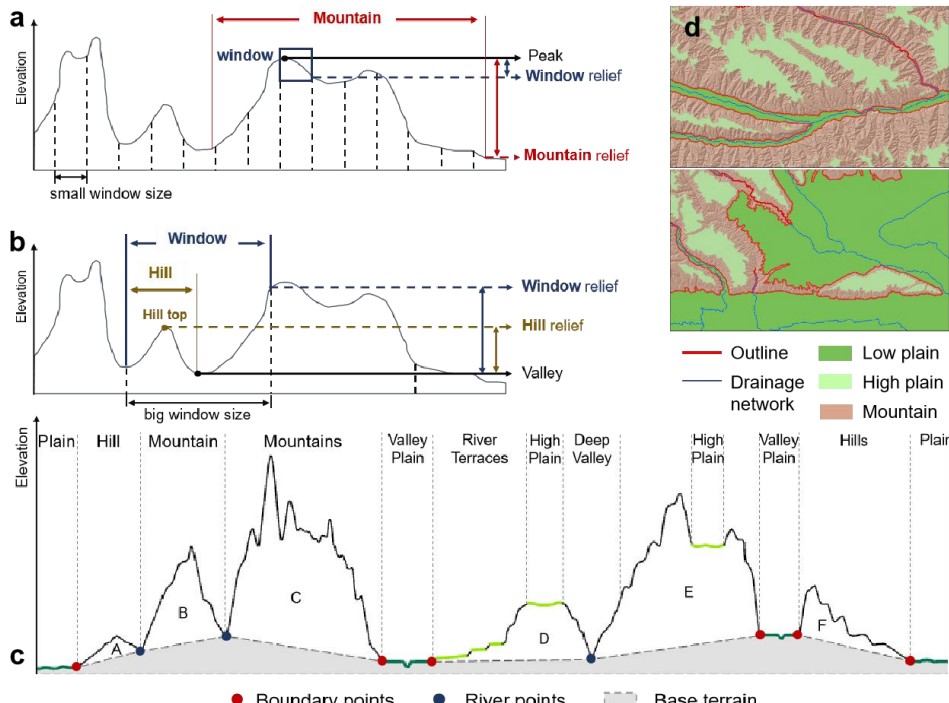


**Figure 3**. **Uncertainty in relief calculation based on the window analysis. a** and **b** the relationship between different windows
and topographical relief. **c** schematic diagram illustrating the base terrain of mountains. **d** features used to create TIN and build base
terrain.
Then, we focus on the differences of terrain relief to achieve the comprehensive classification of L2 landforms. Terrain
relief refers to the difference in elevation between the highest and lowest points within a particular spatial unit. This factor
significantly influences landform classification. However, commonly employed indices reflecting topographic relief are achieved
using a window of fixed size such as 3×3, 5×5 pixels, or larger (Maxwell and Shobe, 2022), a method that fails to account for
geomorphological semantics, and which therefore disregards the integrity of a mountain. Window size has a significant impact on
results of relief calculation. As shown in Figures 3a and b, window analysis tends to disrupt the integrity and continuity of
geomorphological elements. Moreover, a small window size is insufficient to capture the entire mountain, particularly in the case
of large mountains, while a large window size may incorporate other mountains and fail effectively to capture the relief. The
uncertainty introduced by window size further increases the difficulty of global classification and mapping based on relief. Therefore,
we propose a new method for surface relief calculation.
In quantitative analysis, it is crucial to consider the underlying terrain of mountains to accurately assess changes in elevation.
According to the above consideration, we construct mountain units as fundamental analysis units and propose a novel derivative
named the surface uplift index (SUI). In this paper, surface relief is defined as the degree of uplift relative to the flat areas surrounding
the mountains. We regard the elevation at the foot of the mountain as the base elevation and then calculate the elevation difference



between each position on the mountains and the base elevation. Compared to the traditional method of relief calculation (e.g.,
difference in elevation within a particular window size), SUI considers the vertical elevation differences between the surface and
the mountain base, which is more consistent with the human perception of mountain morphology.
This step includes three sub-procedures. Firstly, we constructed the unit 'rugged' and associated fluvial features based on
the boundary of plain. The plain boundary then lies at the foot of landforms classified as mountain. However, when the area of the
mountain is large, and the base elevation is constructed only on basis of the plain boundary, the result may be inaccurate. To refine
the representation of surface relief, we also take into account linear features representing the rivers. These additional lines can be
obtained through DEM based hydro-analysis (Li et al., 2021). In order to ensure that plains at high elevations do not interfere with
the definition of the mountain unit, since these are, in effect, part of the mountain range (Figure 3c) we exclude high elevation plains
that have no fluvial features to retain the integrity of the associated mountain range. Figure 3d shows the elements involved in
establishing the base elevation, which corresponds to the boundary of the low altitude plains and fluvial features (marked in red in
Figure 3d), therefore excluding high altitude plains (marked in light green in Figure 3d). Secondly, we constructed the base elevation
to underpin the calculation of the SUI. In this case, the rugged unit replaces the analysis window. In this step, we constructed the
triangulated irregular network (TIN) based on the position extracted in the first step and then regard these TIN data as the base
elevation. The construction of TIN can be achieve in ArcGIS Pro through create TIN. Thirdly, the SUI is obtained by calculating
the difference between each cell height and its corresponding base elevation. This novel method provides a more appropriate
representation of the underlying terrain.
**2.2.4 Type refinement for L3**
According to the results of previous studies (Zhou et al., 2009), we constructed the classification criteria shown in appendix
of Table A1. For the plains, we use altitudes of 1000m, 3500m and 5000m as break points to generate low-, middle-, high- and
highest-altitude landforms. Mountains are classified as hill, low-relief, middle-relief, high-relief, and highest-relief mountains, based
on threshold SUI values of 200m, 1000m, 3500m and 5000m. In all, this yields 6 L2 and 23 L3classifications.
**2.2.5 Post-processing**
Following completion of the above processes, a map is generated that includes all the basic landform units. However, due
to interference caused by the existence of locally steep changes in topographic relief, this output still contains some features in the
plain areas misclassified as hills. Meanwhile, although the data we used are of high resolution and good quality, outliers and/or data
noise remain. Such anomalies may result in small landform blocks with relatively low terrain relief and, in accommodating this,
we designed an optimization process to correct hill misclassification. We used area and SUI as reflecting their characteristics (e.g.
fragmented and relatively low relief). Considering the application of landform data in geomorphologic mapping and the resolution
of basic data, we determined that our study corresponds approximately to the equivalent of 1:200,000 geomorphological mapping.
Under the conditions of 1:200,000 scale, the minimum displayable patch size is approximately 0.16 km$^2$. The SUI threshold is
derived from (Zhou et al., 2009), which defines plains as the blocks with relief of less than 30 metres. Therefore, blocks with areas



of less than 0.16 km$^2$ and SUIs below 30 metres are regarded as misclassified blocks which are then integrated as part of the
surrounding plains.

221          Meanwhile, we designed an additional step to optimize the results for desert areas. Many arid regions are characterized by

dunes, which are distinctive aeolian landforms of varying shape and size constructed from unconsolidated sand (Hugenholtz et al.,
2012). Dunes are generally smaller in scale than mountains and this challenges our approach to basic landform mapping (Shumack
et al., 2020), increasing the difficulty of accurate dune mapping. In this study, we regarded sand dunes as hills due to their
morphological similarity. However, due to the variation of dune size and shape, it is challenging to correctly classify these dunes as
hills according to our proposed method. Therefore, we design an optimization step to correct the classification results in which dunes
and inter-dune areas are separated and identified according to their altitude and SUI. Firstly, on the basis of on their
geomorphological characteristics, remote sensing images, and hillshade maps, we demarcated the major global sand desert regions.
Secondly, we used the DEM to extract the topographic feature lines by surface analysis of extracting desert feature lines. Employing
the SUI calculation as for other regions, we then constructed the base terrain, in this case, the river networks were extracted with
the threshold $T_{D1}$ of 20000, and then we extracted sampling points from these networks to construct TINs. We calculated the SUI
and then set the segmented threshold $T_{D2}$. Due to inconsistencies in the scale of dunes worldwide, we applied an adjustable $T_{D2}$
ranging from 2m to 10m. Areas less than $T_{D2}$ are defined as inter-dunes (equivalent to plains in the basic landform classification).
All patches smaller than $T_{D3}$ 0.02km$^2$ were regarded as fragments and integrated into the surrounding vector blocks. Finally, we
employed the smoothing tool to ensure appropriateness of the landform boundary.





## 3 Results and discussion

### 3.1 Global landform classification results

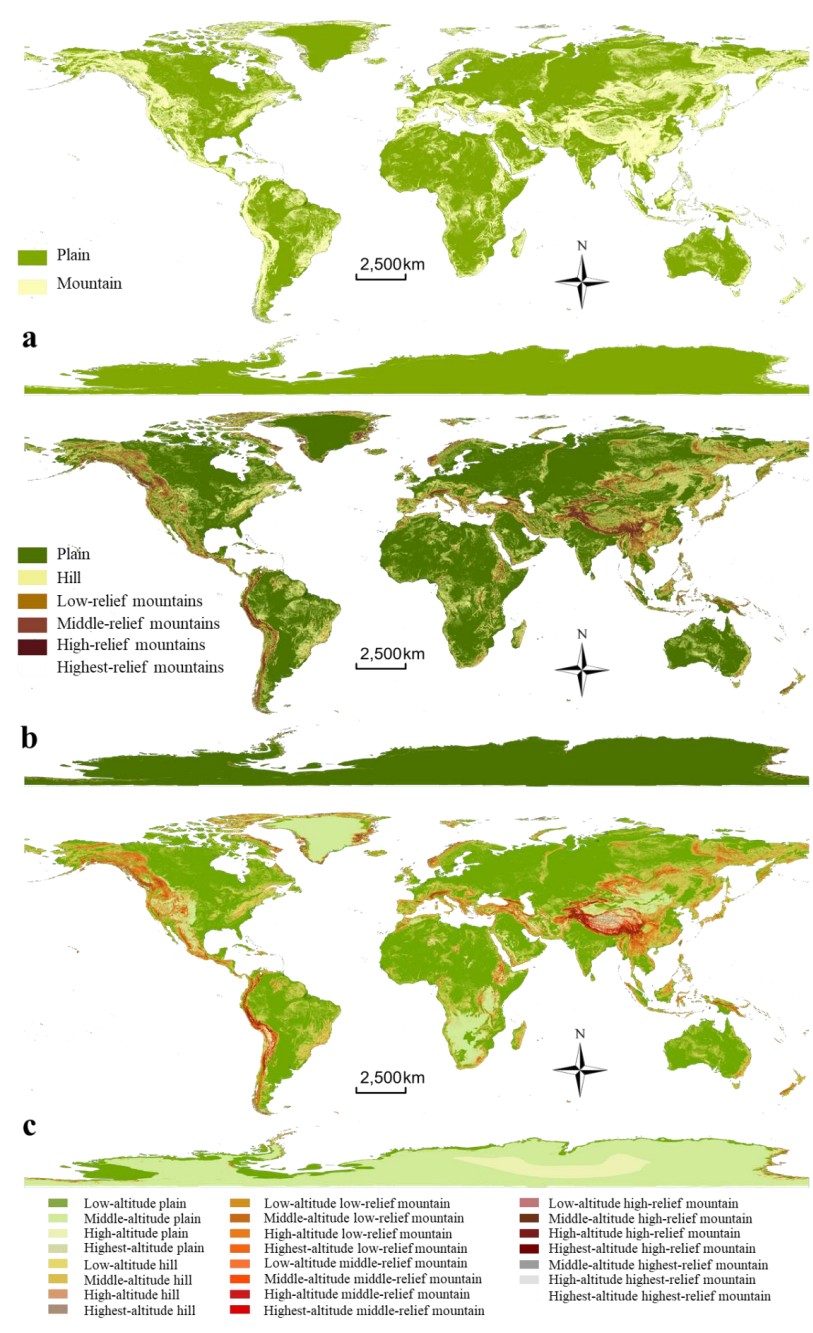

**Figure 4. Results of the basic global landform classification with 30 m resolution.** a, b and c represent the L1, L2 and L3

landforms, respectively.





Figure 4 shows the global landform classification results based on the abovementioned framework. This hierarchical dataset
provides a more comprehensive understanding of the Earth surface. To visualize the results in detail, three typical regions are
selected to demonstrate the performance of the GBLU dataset. Figure 5 shows the GBLU in typical regions and corresponding
remote sensing image from Esri world imagery. The selected regions contain examples of the main landforms on Earth, as well as
transition areas of different landforms. In the mountainous areas as shown in Figure 5a, mountain range and valley orientation is
clearly discernible. The GBLU clearly illustrates the transition zones between mountains and plains, as well as potential floodplains.
While such phenomena are visually discernible in remote sensing imagery, using our proposed framework, they are extracted based
on quantified morphological characteristics. The abundant textural information provided by GBLU can facilitate study of areas with
high geomorphological value, such as fjords (Figure 5b). In desert areas Figure 5c, GBLU effectively illustrates the transitional
patterns between dunes and depressions.  Based on abundant morphological characteristics, GBLU can depict sand dune boundaries
that are strikingly consistent with those visible in imagery. This further underscoring the performance of GBLU in capturing detailed
geomorphic features across varied terrains.

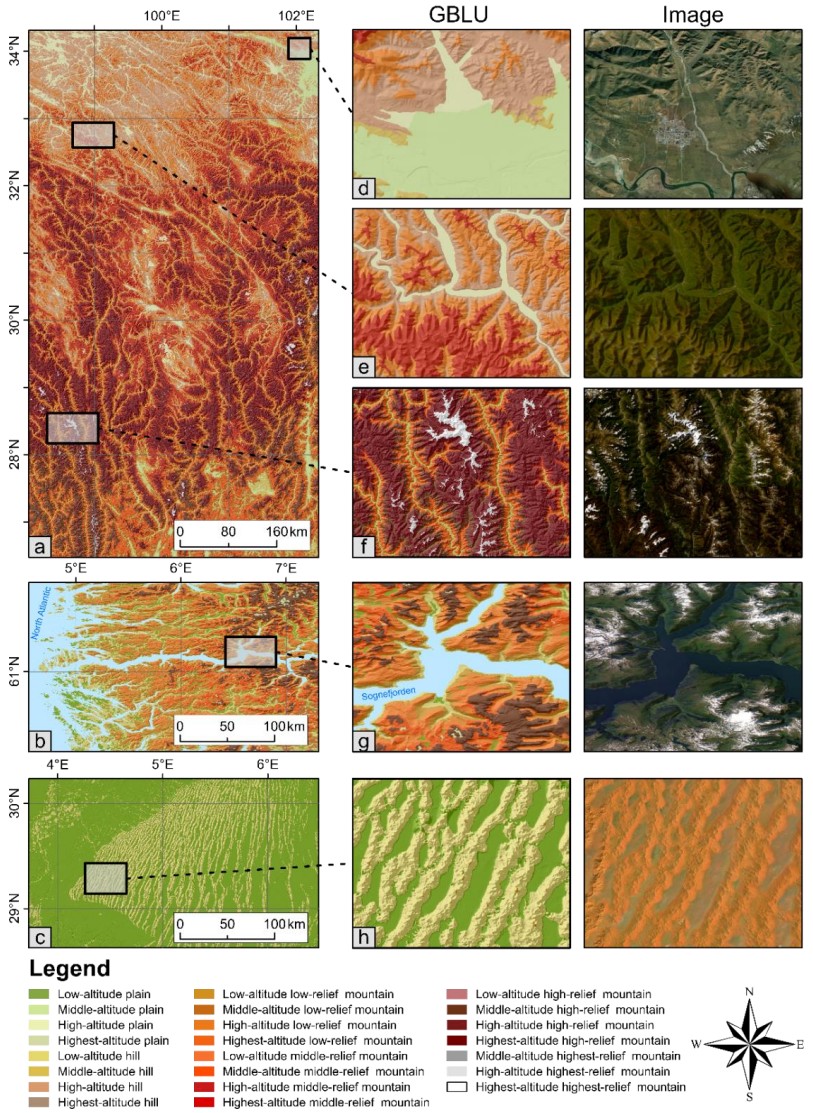

**Figure 5. Comparison of landform classification results and remote sensing imagery.** a eastern part of the Tibetan Plateau. b the Fjord coast in western Norway. c desert area in the central Sahara. e-h are local enlarged areas.

**3.2 Result comparison and validation**

We conducted comparisons between the GBLU dataset and multiple other datasets to comprehensively evaluate our results. Specifically, we compared the outcomes of five landform classifications across a range of sample areas. The most significant improvement achieved by applying GBLU is the increased detail in representing terrain features. The GBLU-based landform classification markedly enhances delineation of independent landforms, such as dunes and mountains, which have clear boundaries and serve as key elements in the analysis of spatial structure and interactions. The classification systems of RefData 3 and 4 are similar to GBLU but have a coarser resolution of 1 km, making them less effective in capturing terrain details. Figure 6 illustrates that there is a variation in the understanding of landform types among different scholars. As stated



by the authors, RefData1 and RefData2 align more closely with terrain classification systems. Although these categories do
include common landforms such as plain and mountains etc., they also encompass other types of terrain features like slope.
In this paper, we consider landforms of plain or mountain to represent larger scales relative to terrain objects like "slope."
Therefore, in designing the classification system, we think that categorizing 'slope' at the same level as 'plain' or 'mountain'
can lead to some comprehension difficulties. Therefore, GBLU offers a more comprehensive landform classification system
and expresses the integrity of landform objects more closely aligned with the ontological understanding of landforms.

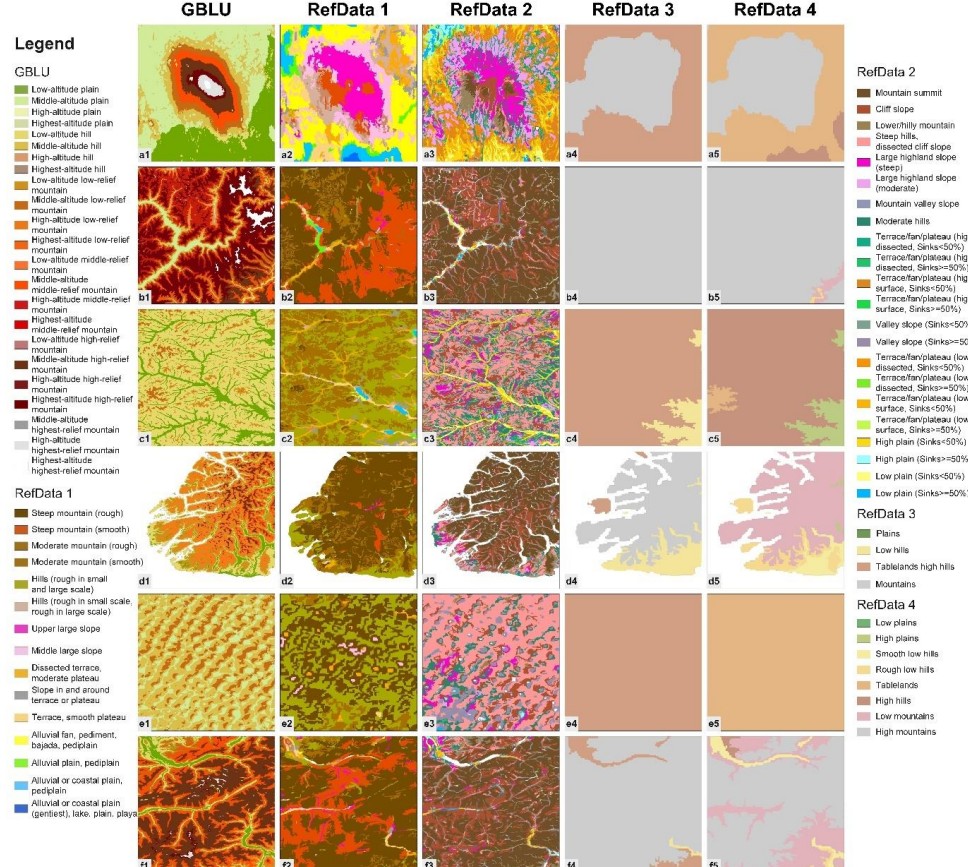


**Figure 6. Comparison of GBLU with RefData 1 - 4.** Selected study areas, from top to bottom, are as follows: a. the Kilimanjaro,
b. Namcha Barwa in Himalaya, c. Greater Khingan Mountains, d. Fjords in New Zealand, e. Badain Jaran Desert and f. Central
Alps. Refdata1 is the 15-class global terrain classification created by Iwahashi et al. (2018) based on 280m DEM. Refdata2 is the
The 22-class global terrain classification created by Iwahashi and Yamazaki (2022) based on 90m DEM. Drăguţ and Eisank's
results include three levels; here we present results of their level 2 (RefData 3) and level 3 (RefData 4) classification.



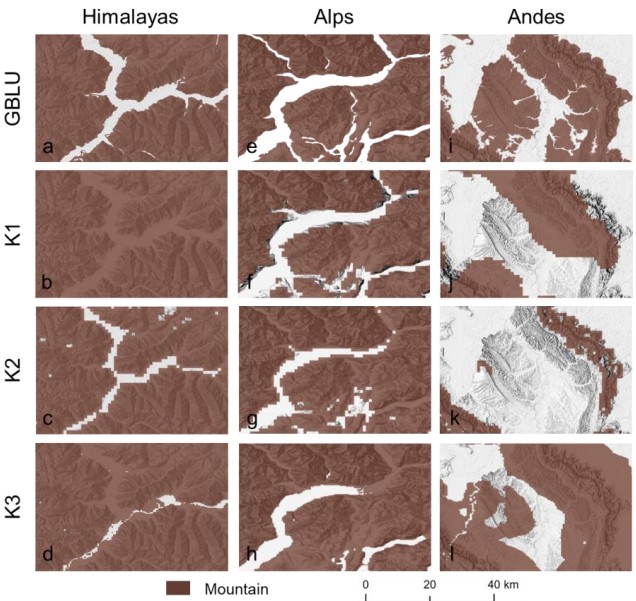


**Figure 7. Comparison between the GBLU and the Global Mountain Biodiversity Assessment (GMBA) projects**.

We conducted a more detailed comparison for mountain regions using the Global Mountain Biodiversity Assessment (GMBA)
(Snethlage et al., 2022) as reference data. The GMBA dataset contains three subsets using the DEM with spatial resolutions of 1000
m, 1000 m and 250 m to generate global mountain maps. These three datasets (e.g., K1, K2 and K3) are produced by analyzing the
morphological derivatives, using a moving neighbourhood analysis window for relief, elevation, and slope (Kapos et al., 2000;
Karagulle et al., 2017; Körner et al., 2011). That similar indicators are used in the associated classification and mapping processes
indicates the comparability of the GMBA and the GBLU datasets, although due to differences in the category settings among the
GBLU and the GMBA datasets, the comparison in this study focused only mountains. As shown in Figure 7, the GBLU dataset
clearly outperforms the other three datasets in depicting mountain details, especially in representing valleys. This can be seen in
Figures 7a-h, whereby the K1,K2 and K3 data exhibits separated upland blocks in mountainous regions with complex and intense
terrain variations, and fails to represent continuous valleys.
Due to differences in classification systems and indices, it is challenging to conduct further quantitative comparisons between
GBLU and other results. To facilitate comparison between these datasets, we merged some classes in the datasets to maintain
classification consistency. For example, we merged mountain summit and cliff slope sections into 'mountain' as per merging criteria
described in Table A2. Overall, GBLU results are consistent with other systems in terms of the macroscopic landform patterns. The
merged results indicate that Iwahashi and Yamazaki's dataset performs better in representing plains boundaries and their shapes.



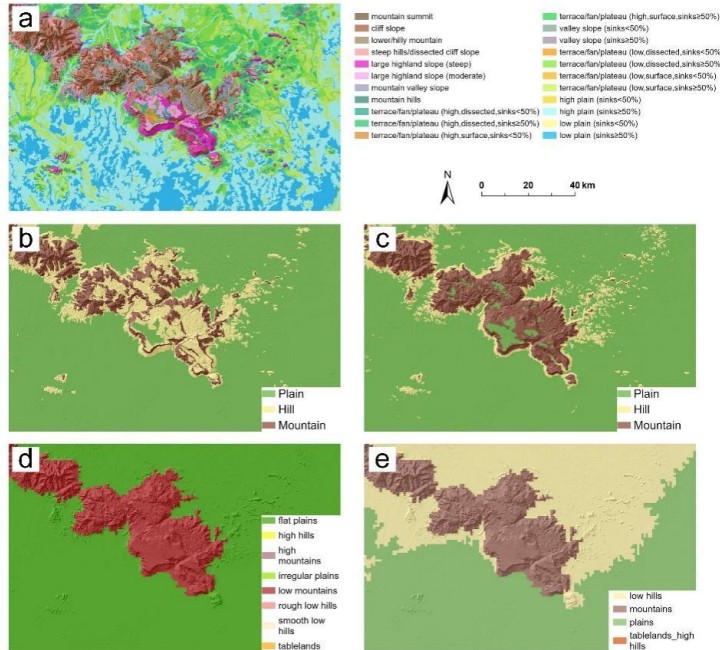


**Figure 8. Classification result of the GBLU for an existing landform mapping dataset in the Amazon River basin.**

**a** Iwahashi and Yamazaki (2022) original result; **b** adjusted Iwahashi and Yamazaki ,2022 result through merging landform
classes; **c** GBLU result; **d** Drăguţ and Eisank (2012) result (level 3); **e** Drăguţ and Eisank, 2012 result (level 2).
**3.4 Global landform composition**

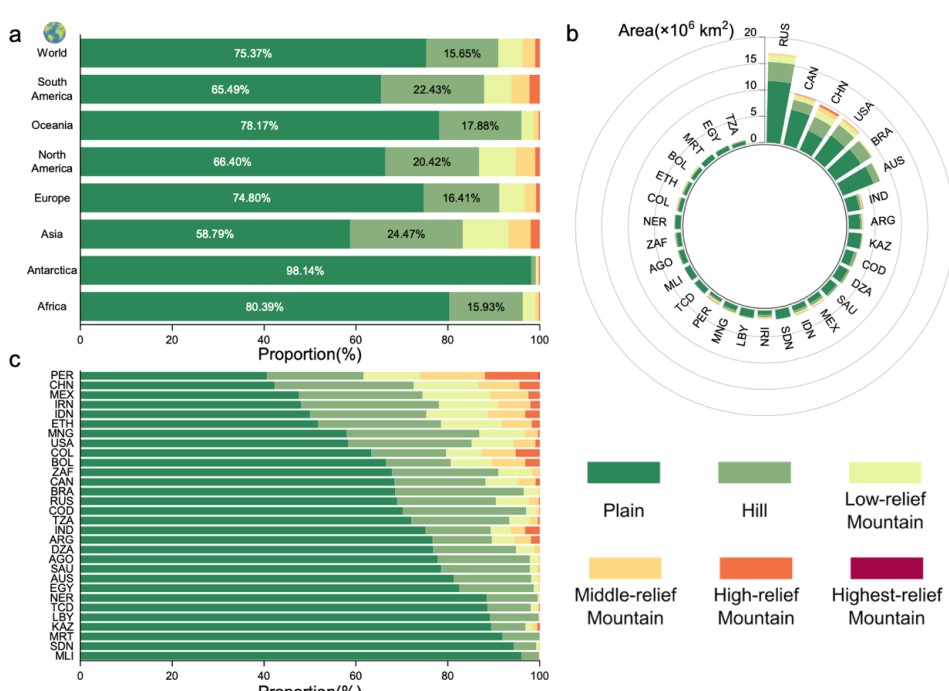






**Figure 9. Area and proportional area statistics at continental and national scales**. **a** Proportion of primary landforms on each
continent. **b** Area of primary landform types in the top 30 countries ranked by area. **c** Proportion of primary landform types in the
top 30 countries ranked by area. Full names of countries listed can be found in Table A3.
We have used a cell size of 500 m x 500 m to accurately assess the proportions of primary landforms across continents
worldwide, thereby yielding insights into their spatial variations. The findings indicate that approximately 75% of the global land
area comprises plains, while some 16% consists of hills, with the remaining portion classified as mountains (Figure 9a). In terms of
the distribution of landform composition, Asia exhibits a very distinctive pattern, since plains cover only 59% of its land area, the
lowest among all continents, while there is a significantly higher proportion of hills and mountains, consistent with its pronounced
topographic diversity. Compared to the global average, the presence of continental marginal mountain chains results in a
significantly lower proportion of plains, and correspondingly higher proportion of mountains, in both North and South America.
Indeed, South America has very substantial areas of high relief mountains, while Africa is distinguished by the dominance of
extensive plains.
We further conducted a comprehensive analysis of landform types and their proportions at the national and regional scale
across all countries and regions worldwide to reveal patterns of variation. Figure 9b illustrates the proportion of primary landform
types in the top 30 countries ranked by area, while Figure 9c depicts the standardized proportion of the landform types within these
countries, sorted based on the proportion of plains. China's diverse and rugged topography is evident in its significantly high
proportion of mountains, while Peru contains the lowest proportion of plains, as mountainous terrain there occupies over 60% of its
land area.
**3.4 Dataset usage note**
In this section, we highlight the results of experiments performed to analyse the relationship between landforms, climate and
land cover to highlight the potential applications of GBLU. Based on the high resolution landform classes provided by GBLU, we
can explore the complex and in-depth relationships between landforms, climate, and land cover. The climate data is the widely used
1-km Köppen-Geiger climate classification maps in 1991–2020 (Beck et al., 2023) and the land cover data is from FROM-GLC
30m in 2017 (Yu et al., 2013).
The enhanced resolution and detail of the GBLU enables subtle variations in the Earth's surface to be captured, which is highly
valuable in understanding interactions between geomorphology and other factors. As shown in Figure 10, landform distribution in
temperate zones suggests a unique blend of climatic conditions and geomorphologic processes, fostering a diverse array of landforms.
In the climatic zones of tropical, arid, and cold regions, we observe that low-altitude plains and hills are most prominent. For polar
areas, a larger proportion of the area is located at higher altitudes than in other climate zones. Regarding land cover analysis
(excluding the South Polar area), cropland occupies 84.27% of plains and 15.73% of mountains, yielding useful insights for
analyzing cultivated land productivity. Meanwhile, forests and bare land are more prevalent in mountains, more especially in hills.
Additionally, the percentage of many ecologically significant biomes, such as forests, grasslands, wetlands, tundra, and water bodies,
in plains and mountainous regions has been brought up to date. This is potentially valuable for assessing the quality of ecological
environments and carbon stocks.

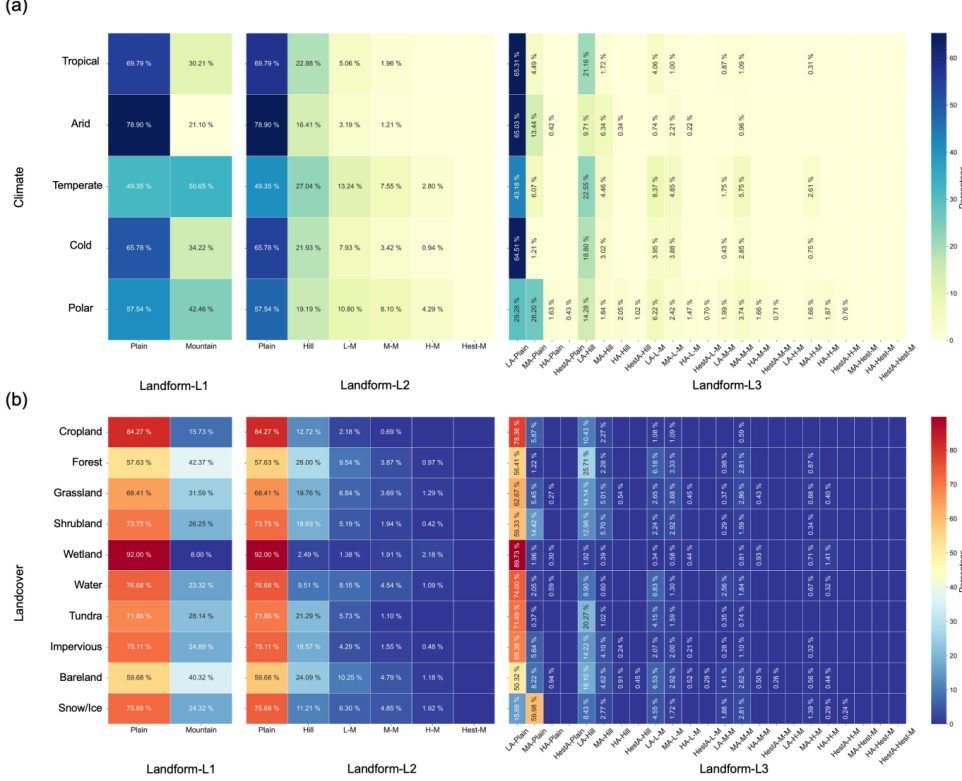


**Figure 10. Relationship of landform types to climate and land cover.** (a) and (b) show the proportions of the three classes of

landform types in different climatic and land cover regions respectively. Values less than 0.2% are not labeled with numbers.
The GBLU provided in this work has obvious applications in geomorphology but also in other fields and can, moreover, play
a fundamental role in supporting the identification of landforms that incorporates complex semantics. For example, identification
of a landscape element as 'tableland' is complex, differs between disciplines, and requires that both morphological and evolutionary
characteristics be accounted for. The GBLU can be integrated with additional observations to map the occurrence and distribution
of tablelands through the delineation of segments that are elevated, flat, and surrounded by steep escarpments. There is also
significant potential for the application of GBLU to other fields (such as geology, hydrology and ecology) focusing on the natural
environment. For example, for ecologists, biodiversity distribution across different landform regions is one of the most significant
issues and central to understanding the nature of ecosystem change. At the regional scale, contrasting geomorphological conditions
are known to promote isolation of biological populations, influencinge community structure and function, as well as evolution.
Meanwhile, the interaction between geomorphology and biogeography may result in complex biogeomorphological dynamics. The
feedback between physical, ecological and evolutionary components constituting biogeomorphological systems is an important



element of the evolution of the Earth's surface.
**4. Dataset access**
Global Basic Landform Units (GBLU v1.0) is stored in the Deep-time Digital Earth Geomorphology platform and Zenodo
(Yang et al., 2024; https://doi.org/10.5281/zenodo.13187969). The data are stored in Esri shapefile format using the coordinate
system WGS84. Total size of the dataset is 150GB, with 6,849,306 independent landform blocks. In order to facilitate application,
we employed a 1° × 1° grid to tile the data for storage, with 25,252 file tiles in all. We distinguish the types of landform units by
coding attributes of the elements. Additionally, we provide a rasterized dataset (at 30m resolution) using the coordinate system of
WGS84. Values of the cells represent the codes of L3 types. In the attribute table, field "code0" is the landform type code of the
first level, field "code1" is the landform type code of the second level and field "code2" is the landform type code of the L3.
**5. Conclusion**
This study provides a novel global landform classification dataset (GBLU) with a resolution of 1 arc-second
(approximately 30 m). In this study, we propose a novel framework for global landform mapping to significantly improve
the quantitative evaluation of geomorphological features. The key output is the release of the GBLU dataset that is suited to
applications across multiple disciplines, including geography, geology, ecology, and hydrology. Global-scale analysis of attributes
within the GBLU reveals the composition and distribution of global landforms that enables comparison between regions and
continents. The results emphasize the notable heterogeneity of Asia in general, and of China in particular, in terms of
geomorphological diversity. The GBLU outperforms previous datasets in expressing landform details, providing an
opportunity to investigate the Earth's natural resources. The resolution of the GBLU is similar to that of the current
mainstream remote sensing data, which makes combined use of the data relatively simple. We believe that this dataset
can provide abundant and detailed geomorphological information for the field of earth sciences, facilitating further
advancements in related research.



**Appendix A**

**Table A1.** Classification of global basic landform types

| L1 | Code | Colors (RGB) | L2 | Code | Colors (RGB) | L3 | Code | Colors (RGB) |
|---|---|---|---|---|---|---|---|---|
| Plain | 1 | 129,168,0 | Plain | 11 | 76,115,0 | Low-altitude plain | 111 | 112,168,0 |
| | | | | | | Middle-altitude plain | 112 | 209,235,152 |
| | | | | | | High-altitude plain | 113 | 237,242,179 |
| | | | | | | Highest-altitude plain | 114 | 213,217,164 |
| Mountain | 2 | 255,255,190 | Hill | 21 | 240,242,148 | Low-altitude hill | 211 | 230,216,106 |
| | | | | | | Middle-altitude hill | 212 | 220,191,75 |
| | | | | | | High-altitude hill | 213 | 217,155,110 |
| | | | | | | Highest-altitude hill | 214 | 170,141,117 |
| | | | Low-relief Mountain | 22 | 168,112,0 | Low-altitude low-relief mountain | 221 | 209,145.28 |
| | | | | | | Middle-altitude low-relief mountain | 222 | 198,106,20 |
| | | | | | | High-altitude low-relief mountain | 223 | 237,122,24 |
| | | | | | | Highest-altitude low-relief mountain | 224 | 244,100,18 |
| | | | Middle-relief Mountain | 23 | 137,65,47 | Low-altitude middle-relief mountain | 231 | 253,120,25 |
| | | | | | | Middle-altitude middle-relief mountain | 232 | 255,76,0 |
| | | | | | | High-altitude middle-relief mountain | 233 | 201,30,9 |
| | | | | | | Highest-altitude middle-relief mountain | 234 | 220,0,0 |
| | | | High-relief Mountain | 24 | 86,20,24 | Low-altitude high-relief mountain | 241 | 193,119,120 |
| | | | | | | Middle-altitude high-relief mountain | 242 | 110,50,20 |
| | | | | | | High-altitude high-relief mountain | 243 | 114,4,9 |
| | | | | | | Highest-altitude high-relief mountain | 244 | 115,0,0 |
| | | | Highest-relief Mountain | 25 | 255,255,255 | Middle-altitude highest-relief mountain | 252 | 156,156,156 |
| | | | | | | High-altitude highest-relief mountain | 253 | 225,225,225 |
| | | | | | | Highest-altitude highest-relief mountain | 254 | 255,255,255 |



**Table A2**. Merging the GBLU results to enable comparison with the results of Iwahashi and Yamazaki.

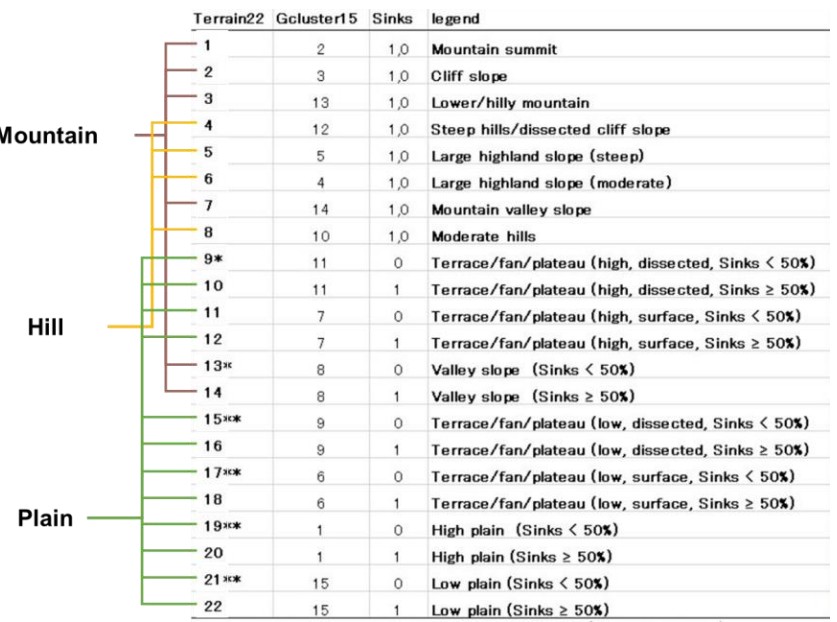

**Table A3**. Countries' names and their abbreviations.

| NAME | Abbreviations |
|---|---|
| Russian Federation | RUS |
| Canada | CAN |
| Peoples Republic of China | CHN |
| United States of America | USA |
| Federative Republic of Brazil | BRA |
| Commonwealth of Australia | AUS |
| Republic of India | IND |
| Argentina | ARG |
| Republic of Kazakhstan | KAZ |
| Democratic Republic of Congo | COD |
| Democratic People | DZA |
| Kingdom of Saudi Arabia | SAU |
| United States of Mexico | MEX |
| Republic of Indonesia | IDN |
| Republic of the Sudan | SDN |
| Islamic Republic of Iran | IRN |
| Great Socialist People | LBY |
| Mongolia | MNG |
| Republic of Peru | PER |
| Republic of Chad | TCD |
| Republic of Mali | MLI |
| Angola | AGO |
| Republic of South Africa | ZAF |
| Republic of Niger | NER |
| Republic of Colombia | COL |
| Federal Democratic Republic of Ethiopia | ETH |
| Republic of Bolivia | BOL |
| Islamic Republic of Mauritania | MRT |
| Arab Republic of Egypt | EGY |
| United Republic of Tanzania | TZA |



## Author contribution

Xin Yang, Guoan Tang and Michael Meadows  designed the study.
Xin Yang, Sijin Li, Junfei Ma, Yang Chen and Xingyu Zhou performed the analysis.
Xin Yang and Sijin Li wrote the first version of the manuscript.
Fayuan Li, Liyang Xiong and Chenghu Zhou coordinated the work and reviewed the manuscript.
Sijin Li, Junfei Ma, Yang Chen and Xingyu Zhou  assisted with quality control and reviewed the manuscript.
All the authors contributed to the final version of the manuscript.

## Competing interests

The authors declare that they have no conflict of interest.

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
