# Peer review of "A typology of global relief classes derived from digital elevation models"

_Earth System Science Data, 2024_

## Author Comment (AC1)

The authors present a global dataset of landforms derived from a high-resolution DEM. They propose new ways to identify plain areas and their transition to hilly and mountainous terrain. The novel way to do this by identifying core areas and including transition areas through a cost distance analysis yields results that seem visually quite accurate when the map is overlayed onto a relief background. Plain and higher relief areas are neatly differentiated. This type of information can be quite useful for geographical and ecological macro studies. The precise workflow does miss details to be reproducible. It is a pity that proprietary software was used and the workflow described in general terms only, which makes replication more difficult. The choice for some cut off values or thresholds (slope, elevation, accumulated cost) is not always clearly explained or motivated.

**Response:**

Thank you for your valuable feedback. We appreciate your recognition of the dataset's potential for geographical and ecological macro studies.

We acknowledge the need for a more detailed workflow description to enhance reproducibility. In the revised version, we have provided additional details on the specific steps, including explanations of the principles guiding our selection of threshold values (slope, elevation, and accumulated cost).

To classify the hilly and mountainous areas the authors propose a new approach as an alternative to a moving windows analysis that has documented limitations. Landform relief is not calculated with reference to the nearest elevation data within a (small) window, but expressed with reference to a regional baseline calculated by creating a TIN on the basis of the elevation at the border of a mountain range (i.e. where it transitions to plain). In addition, the baseline elevation takes into account the elevation of points along water courses within the mountain to create a baseline surface to act as reference for the roughness calculations. Thresholds are applied to the elevation differences calculated by subtracting the baseline elevation from the actual surface elevation. The lowest elevation differences are labelled hills, followed by low relief mountain up to highest relief mountain. This leads to a conceptual problem. In my opinion, when one talks about a mountain or mountain range such as the Himalaya as a landform, one considers the mountain as a whole, from the foothills to the highest summits as the landform "highest relief mountain". Similarly when talking about the Jura or the Vosges mountains, one would talk about low relief mountains, but not consider only the mountain summits to be low, but the whole landform down to the foot slopes as being the low mountain.

**Response**:

Thank you for your valuable feedback. We carefully reviewed your concern and analyzed the potential reasons behind the differences in defining what constitutes a mountain. We think that the perception of a mountain is largely scale- and context-dependent. Below, we present two key considerations:

**(1) Scale Perspective**: In general geomorphometry, "landform" can refer to units at multiple scales. In our results, "mountain" in Level 1 (L1) aligns with the conventional, broader concept of a complete mountain entity. Levels 2 and 3 (L2 and L3) aim to capture local variations within that mountain by subdividing it according to specific altitude and relief thresholds. In other words, Levels 2 and 3 represent finer-scale morphological facets compared to the L1 "mountain." To avoid confusion about terms, we have renamed the L3 'mountain' classes to 'mountain slope', thereby clarifying that L3 focuses on local slope-based subdivisions rather than a single, unified mountain. This multi-scale approach allows users who only need a macroscopic view (i.e., one label for the entire mountain range) to rely on L1. Meanwhile, researchers focusing on localized processes (e.g., slope erosion, microclimate differences, or altitudinal ecological zones) may benefit from the finer distinctions at Levels 2 and 3.

**(2) Context Perspective**: As you noted, viewing a mountain as a single entity is a common perspective, emphasizing its unified formation process and general independence. In our study, L1 was designed to capture this "common landform" notion of a mountain. In GBLU, we have improved the bounding precision of L1 with higher-resolution data and advanced methods. However, "mountain" can be a somewhat vague term—different definitions may be useful for different contexts [1]. Mountains often exhibit significant internal variability in altitude, relief, and slope, which in turn can influence climate, vegetation, biodiversity, and geomorphic processes. Because the GBLU dataset is intended for broad usage in geoscience, L2 and 3 highlight these internal subdivisions, which is particularly relevant for analyses of force accumulation, mountain ecosystems, and microclimatic gradients. Similar approaches are reported in the subfields of geoscience such as climate, ecology and biology [2-5]. From this perspective, subdividing what is commonly called "a single mountain" into multiple levels is necessary in many research scenarios.

In the revised manuscript, we have supplemented the text with more details on how each level's terminology is constrained to avoid ambiguity (Lines 99-106). Meanwhile, we have updated naming conventions within Level 3. Specially, Level 3 classes initially labeled as "mountain" have been renamed to

"mountain slope" to reflect their smaller-scale morphological nature. We hope these clarifications address your concerns.

[1] Evans, I.S., 2012. Geomorphometry and landform mapping: What is a landform?. Geomorphology, 137(1), pp.94-106.

[2] Antonelli, A., Kissling, W.D., Flantua, S.G., Bermúdez, M.A., Mulch, A., Muellner-Riehl, A.N., Kreft, H., Linder, H.P., Badgley, C., Fjeldså, J. and Fritz, S.A., 2018. Geological and climatic influences on mountain biodiversity. Nature geoscience, 11(10), pp.718-725.

[3] García-Ruiz, J.M., Arnáez, J., Lasanta, T., Nadal-Romero, E. and López-Moreno, J.I., 2024. The Main Features of Mountain Vegetation and Its Altitudinal Organization. The Timberline. In Mountain Environments: Changes and Impacts: Natural Landscapes and Human Adaptations to Diversity (pp. 167-202). Cham: Springer Nature Switzerland.

[4] Rahbek, C., Borregaard, M.K., Antonelli, A., Colwell, R.K., Holt, B.G., Nogues-Bravo, D., Rasmussen, C.M., Richardson, K., Rosing, M.T., Whittaker, R.J. and Fjeldså, J., 2019. Building mountain biodiversity: Geological and evolutionary processes. Science, 365(6458), pp.1114-1119.

[5] Rahbek, C., Borregaard, M.K., Colwell, R.K., Dalsgaard, B.O., Holt, B.G., Morueta-Holme, N., Nogues-Bravo, D., Whittaker, R.J. and Fjeldså, J., 2019. Humboldt's enigma: What causes global patterns of mountain biodiversity?. Science, 365(6458), pp.1108-1113.

In some cases the transitions from different categories of mountain to hilly land is well captured in this approach, typically in ancient eroded landscapes with remnants of higher mountains. The dissected rolling hill landscape gets the label hills, while the remaining inselbergs are classified as mountain.

However, the story is very different in younger mountain areas such as the European Alps or Himalayas. If one looks at the GBLU map without legend overlaid onto a relief map, valley-like shapes appear very distinctly that follow the actual valleys of these mountains. When looking at the legend, one sees that these are actually classified as hills. The same holds for flat valley bottoms inside the mountains, these are classified plains, even if they are long, narrow and sinuous.

In my conceptualization of a mountain, the mid slopes of high mountains do not pertain to the landform class middle relief mountain. They are mid slopes of a high mountain. Similarly, mountain valleys are not hills, just because the local surface elevation is below a certain threshold.

**Response:**

Thank you for your detailed analysis of our results, particularly regarding ancient eroded landscapes. We appreciate your in-depth perspective, and in response to the issues you raised, our considerations are as follows:

First, to avoid semantic confusion, we have renamed all Level 3 "mountain" to "mountain slope". This change clarifies that Level 3 targets finer-scale morphological units rather than large, unified mountain bodies.

Secondly, we fully acknowledge the importance of valleys in geomorphological research. One major challenge in classifying valleys lies in the absence of a unified definition, particularly regarding the valley extent or boundary on the mountainous slope. This ambiguity is often greater than that for mountains. Moreover, "valley" and "mountain" highlight fundamentally different concepts—one emphasizes downward incision, and the other highlights upward uplift. Thus, if we were to include both "valley" and "mountain" within a single classification system, we would need to define a conceptual interface to separate them. However, there is no broadly accepted standard for doing so. From a technical standpoint, it is also difficult to classify valleys because they typically lack a pronounced terrain break, which complicates classification in traditional approaches.

Although GBLU does not explicitly label "valleys," it does provide a basis for valley extraction. As you noted, the polygons in GBLU—be they categorized as "hill" or "plain"—often capture the shapes of valleys. Our approach captures the cumulative characteristics of landform objects and uses slope accumulation to delineate subunits within a mountain. Thus, even though "valley" is not designated as a distinct category, the GBLU already produces polygons that effectively represent valley-like features. Once a user identifies a specific mountainous region of interest, they can extract those GBLU patches with valley-like shapes (classified as plain or hill, for instance) and reassign them as "valley," thereby defining the valley object according to their own study's requirements.

When I look at the methods and results of this paper, I think of the product as something like "Map of relief classes and relative (or regional) elevation zones", and I am convinced that this classification is useful for different scientific applications. Ontologically I don't think that the presented map units should be thought of as representations of landforms.

In summary, I commend the authors for what seems to be a very detailed and precise work and the product and the work that has gone into its production. Also, the results seem to be useful for certain research applications. I do not

however agree with the authors that what is represented here are landforms, ontologically speaking.

The distinction I make here is further illustrated in the figure.

[Figure]

Figure: Upper transect: how I understand the current version of the GLBU. Lower transect: how I think landforms should be conceptualized in this context.

**Response:**

Thank you for acknowledging the potential applications of our dataset. As noted in our earlier reply, the term "landform" carries multiple meanings, and its specific interpretation depends closely on the chosen scale and disciplinary context. In terms of scale, Levels 2 and 3 in our classification indeed differ from the broader notion of a mountain (i.e., Level 1), yet they all reflect the underlying morphology of the Earth's surface. To avoid semantic confusion, we have renamed all Level 3 "mountain" categories to "mountain slope", thus distinguishing them from the more general, higher-level concept of a mountain. From an application standpoint, given that geomorphology intersects with subfields of geoscience such as climate and ecology, we provide a conventional categorization (i.e., the plain and mountain) at Level 1 while also offering finer distinctions at Levels 2 and 3 to meet more specialized research needs. For further details, please refer to above response.

Regarding the data availability, the authors have presented the resources they developed on Zenodo. The files are easily accessible and useable in open source software. Files are presented in folders by 10 degree latitudinal bands, and it is quite easy to find a region of interest. All terrestrial areas of the world seem to be included in the data. There is a possible issue for global level use

of the data in that it consists of many different tiles that need to be mosaiced, but this can be coded.

**Response:**

Thank you for your suggestion. In this revised version, we have improved the file organization by mosaicking the tiles into 10° × 10° regions. These mosaicked files are now grouped into folders based on their latitude for easier access and use.

The validation is done against a number of similar products where one of the main differences is the resolution of the source layers (DEM), this product being based on very high resolution sources (~30 m at the equator.) The identification of plains seems to be more accurate than in any of the products with which it is compared.

Response:

Thank you for your comment. As you mentioned, more accurate plain boundaries are an important improvement of our dataset, thanks to the innovative approach employed in our methodology.

Overall the manuscript is sufficiently concise, the language clear, although it could benefit from some minor edits here and there (see below). On several points the methods section should be developed a bit further to allow full replication of the work flow.

Overall the language is clear and very understandable, but some suggestions for minor improvements are given.

**Response:**

We appreciate your feedback and have carefully revised the manuscript. We have expanded the methods section to provide more details for the replication (Lines 159-186 and Lines 214-235), and build a code repository to publish our workflow. More details can be found in https://github.com/nnu-dta/GBLU-code.

As said, in my opinion the layers presented in this work do not represent landforms. However I think that the classification of relief in plains and mountains with different values of elevation and relief intensity (roughness) can be quite useful for a series of environmental applications. My recommendation would therefore to revise the title and some sections of the text where the product is labelled as a map of landforms and replace this with formulations that more accurately reflect what is shown, that is, not to speak of landforms but about a map of relief (roughness) and elevation classes (or something

similar) instead. This would require rather limited changes to the text and figures.

**Response:**

Thank you for your suggestion. As noted in our previous responses, landforms have inherent scale and context dependencies. To avoid misunderstanding, we have clearly defined the concepts and scale limitations of the landform types discussed in this study within the manuscript. We have substituted the term "landform" with "relief" or "elevation". Additionally, we have revised some category names in our classification system; specifically, we changed "mountain" at L3 to "mountain slope" to better reflect its terrain-related implications. To ensure comprehensibility in both general geomorphometry and related specific fields, we have retained the term "landform" in the title and in certain sections of the text.

Specific comments

35-36: I would add evolution or genesis to this list of research subfields of geomorphology

**Response:** Thank you for your suggestion. We have added "genesis" to this list (Line 35).

43: I would add that field work is an essential component of landform mapping (geomorphology)

**Response:** Thank you for your suggestion. We have added "the survey based on the field work" in this sentence. (Line 43)

46-47: there is a more recent product produced by Amatulli et al. that might be useful to refer to here: Amatulli, G., McInerney, D., Sethi, T., Strobl, P., & Domisch, S. (2020). Geomorpho90m, empirical evaluation and accuracy assessment of global high-resolution geomorphometric layers. Scientific Data, 7(1), 162. https://doi.org/10.1038/s41597-020-0479-6

**Response:** We have added this reference as your suggestion. (Line 47)

56-58: However, as the authors stated, unsupervised classification based methods to perform higher-resolution global landform classification require an international team with knowledge of geomorphological development in a variety of climatic and physiographic settings. > do you address this?

**Response:** Thank you for your question. In fact, we cannot fully resolve this issue. We included this statement because, as Iwahashi (Iwahashi and Yamazaki, 2022) described, unsupervised methods (such as clustering) require

considerable effort to determine the geomorphological meaning of each category. This is challenging since these derived classes may differ significantly from conventional landform perception. In our study, we hope to optimize this process by pre-defining the landform classification system before applying our technical methods, and we based this system on a comprehensive review of existing work.

69-70: not clear if this paper only object is to classify the shape or also something about the material (lithology) and / or genesis, / evolution. Methods and final product seem to be focusing on shape irrespective of material / genesis.

**Response:** This study focuses primarily on the fundamental morphology of landforms. We have changed them to "maintaining the morphological integrity of the identified landforms" and "diverse and complex environmental factors have shaped different types of increase the complexity landform morphology". (Lines 70-71)

80: objective: "to construct a global classification system for landforms that integrates geomorphological knowledge," : not clear where the geomorphological knowledge comes in in the method

**Response:** After careful consideration, we think that the term "knowledge" could potentially cause misunderstandings. Therefore, in the revised manuscript, we replaced it with "a global classification system for landforms that integrates domain consideration of landform-related studies". (Line 82)

82: typo: "high-resolutiojn" > high-resolution

**Response:** Thank you for your comment. Based on your suggestions and those of the other reviewers, to avoid misunderstanding regarding "high-resolution," we have revised the sentence to:"(4) to make available a comprehensive global dataset of landform units." (Line 84)

99-100 "The first-level (L1) types are defined as 'plain' and 'mountain', reflecting the most fundamental morphological characteristics of landforms." If I understand it well, the first level distinguishes between plain and non-plain (i.e. hills and mountains), as all that is not plain is later subdivided into several classes of hills and mountains, not mountains alone.

**Response:** Thank you for your comment. We have given considerable thought to this naming. In some cases, as you mentioned, hills formed by the erosion

of ancient mountains can be regarded as a subclass of mountains. Therefore, to better capture the general concept of landforms at Level 1, we have retained both category names in L1.

102: "This classification perspective aids researchers in conducting macro-scale studies"  This is indeed a valuable distinction
**Response:** Thank you for your recognition.

113: "the area the missing from FABDEM" >  the area missing from FABDEM
**Response:** Thank you for pointing that out. We have corrected it to "the area missing from FABDEM."

120:  "The following sections provide details that should allow users to reproduce our results."  : some more details would be needed to achieve this I think
**Response:** Thank you for your comment. We have supplemented the manuscript with detailed computational information: (1) we have added the rationale for constructing the accumulated cost (AS) and provided a detailed computational process (Lines 159-186); (2) we have included the detailed calculation process for the new relief metric (Line 214-235).

123: Fig 1:  "accumulate slope " > accumulated slope?
 "Interecting with flat landforms"  > Intersecting with flat landforms
 "Eliminating fragement blocks"  > Eliminating fragment blocks
**Response:** We have changed this figure as your suggestion.

125: data preprocessing or data pre-processing (see figure, perhaps harmonize?)
**Response:** We have removed the hyphen ("-") in the section title to ensure consistency throughout the manuscript.

130:  "data from latitudes below 70° are transposed onto the Behrmann projection, and the remaining data are transported onto the Lambert azimuth equal-area projection. " : suggested edit: Tiles between 70° N/S are reprojected to the equal area Behrmann projection, and the tiles polewards of 70° N/S to Lambert azimuthal equal-area.
**Response:** Thank you for your suggestion. We have replaced the original text with the revised version as you suggested. (Lines 136-141)

132-133: this first sentence is more of a statement that would perhaps be better in the introduction. Starting this section with the second sentence works quite well.

**Response:** We have removed the first sentence and now begin the section with the second sentence for improved flow and clarity.

140: Fig 2b  typo: "varient" > variant

**Response:** We have changed this figure as your suggestion.

147: how large must the continuous area of plain be to be considered a core area? I.e. how many contiguous pixels constitute a plain core area? Do you also apply a shape criterion, or can a very long area of contiguous plain pixels also constitute a core plain area?

**Response:** Thank you for your question. An area must be greater than 0.1 km², and we do not apply any shape criterion. In practice, due to slope limitations—especially in mountainous regions—it is rare to include plain core areas with an extremely elongated shape. We have added explanation in the revised manuscript. (Lines 159-161)

148-150: it is not clear to me what the cost layer is in this calculation: elevation, slope, or something else? Same holds for 'cost' in Fig 2a.

**Response:** In our calculation, the cost layer represents the slope layer. We have clarified this in the manuscript Line 181. Additionally, we have updated the description in Figure 2a to explicitly state that "cost" refers to slope, improving clarity for readers.

149:  "The AS is calculated as the minimum cumulative cost of each position to the nearest landform core along a specific path"  Would it not be more precise to say: The AS is calculated as the minimum cumulative cost of each position to the nearest plain core along a specific path.

**Response:** We have modified the sentence as your suggestion. (Line 182)

155-156: not clear to me how such an algorithm achieves the most direct integration of geomorphological knowledge and expertise

**Response:** Thank you for your suggestion. We have modified the sentence to: "Segmenting landforms through the determination of the thresholds for landform derivatives is one of the most common methods used in

geomorphological studies and transforms geomorphological qualitative perception towards quantitative computation." (Lines 187-188)

160: does T2 have a dimension and a unit? 1500-2000, is that length in meters, or slope in degrees or something else?

**Response:** Thank you for your question. T2 is measured in degree·meters (°·m), representing the accumulated cost-distance where slope (degrees) serves as the cost factor and distance (meters) accumulates along the path.

161-162: "but needs to be determined by integration with expert knowledge within different geomorphic regions". Not clear if you state that this should be done or that it has been done, and if so how?

**Response:** Thank you for your comment. We have revised the statement as follows: "This threshold range is provided as a reference but gentle adjustments to the thresholds may be required in some special areas, such as small islands, through human-computer interaction." (Lines 193-194)

162: "In some cases, it may exceed the recommended threshold range." – not clear where and when

**Response:** Thank you for your question. The statement refers to specific cases where terrain complexity makes it challenging to apply standard threshold values. By referencing hillshade data and satellite imagery, we identified special terrain structures, including small islands, where traditional watershed and TIN-based methods struggle to perform effectively within predefined threshold ranges. We have added explanation in the revised manuscript. (Line 195)

165-167: "This novel method avoids the negative effect of local window analysis and is beneficial for maintaining the landform semantics for each block." Visual inspection of a number of tiles indeed shows a neat identification of the borders of plains and their transition to hilly or mountainous terrain.

**Response:** We appreciate your recognition of how our method effectively delineates the boundaries between plains and transition zones.

176-177: "a method that fails to account for geomorphological semantics, and which therefore disregards the integrity of a mountain. " I would argue that the classification of L2 landforms proposed in this paper does just that. I do not see any landform concept reflected in the classes, and even less so in the map units corresponding to these classes. See general comments above

**Response:** Thank you for your comment. We have provided detailed responses in the general comment section; please refer to that for further information.

192: "on basis of the plain boundary" > on the basis of the plain boundary
**Response:** We have corrected "on basis of the plain boundary" to "on the basis of the plain boundary."

192-193: "To refine the representation of surface relief, we also take into account linear features representing the rivers. " I suppose you do not consider all rivers and streams to construct your TIN of mountain base. Rivers and streams go up to great altitudes. Which sections of mountain rivers did you consider to construct the TIN?
**Response:** In this step, we employed the hydrologic analysis workflow from digital terrain analysis to extract the drainage network. We did not include all rivers or streams; instead, we retained only those of relatively higher order, such as primary or secondary channels. Specifically, we established a segmentation threshold based on flow accumulation—only river networks with values above this threshold were preserved. For reference, in an 11°×11° area, we set a threshold of 200,000, and we adjust this value in accordance with local geomorphic features. For example, in areas with more valleys, the threshold is increased. Regardless of these adjustments, as you mentioned, the final extracted river network does not extend to higher elevation areas.

206: was there any reasoning behind the selection of these elevation bands? 0-1000, 1000-3500, 3500-5000 and >5000?
**Response:** The selection of these elevation bands (0–1000 m, 1000–3500 m, 3500–5000 m, and >5000 m) was based on previous studies, particularly those by Zhou et al. and research on European landscapes. These elevation thresholds reflect major geomorphic and climatic transitions and were chosen to ensure a meaningful classification of landforms based on both process-based and regional geomorphic considerations. Detail information are as follows:
0–1000 m: Represents regions primarily influenced by fluvial erosion, where river dynamics play a dominant role in shaping the landscape.
1000–3500 m: Corresponds to the corrosion function line, a threshold that marks significant shifts in geomorphic processes.
3500–5000 m: Represents areas where periglacial and high-altitude processes become more dominant.

>5000 m: Aligns with the average elevation of modern glaciers, where glacial processes are the primary drivers of landform development.

207-208: idem
**Response:** We carefully reviewed this sentence, but we were unable to fully follow your comment.

277: Figure 7. Comparison between the GBLU and the Global Mountain Biodiversity Assessment (GMBA) projects. > Figure 7. Comparison between the GBLU and three mountain definitions presented on the Global Mountain Explorer (https://rmgsc.cr.usgs.gov/gme/)
**Response:** Thank you for the reminding. We have corrected the figure caption as you suggested.

278-279: this does not seem to be entirely accurate: "We conducted a more detailed comparison for mountain regions using the Global Mountain Biodiversity Assessment (GMBA) (Snethlage et al., 2022) as reference data." The three definitions are from three different institutions (WCMC, GMBA and USGS) but have conveniently been presented together on the Global Mountain Explorer (https://rmgsc.cr.usgs.gov/gme/). The latest mountain definition is the one by Snethlage et al (2002) which can be obtained from https://www.earthenv.org/mountains (scroll down to: Download the GMBA Mountain Definition v.2 here.)
**Response:** Thank you. We have corrected it in the revised manuscript.

337: "fundamental role in supporting the identification of landforms that incorporates complex semantics." > not clear what semantics means in this context
**Response:** In this section, our aim is to emphasize some background knowledge from various specific studies. We changed "complex semantics" to "domain background" in the revised manuscript. (Line 371)

344 "influencinge community structure and function," > influencing community structure and function,
**Response:** We have corrected it.

---

## Author Comment (AC2)

The authors introduce a global landform classification dataset (GBLU) that represents a significant advancement in resolution compared to existing global geomorphological data. Their three-levels classification system with 26 distinct landform classes demonstrates an approach to categorizing Earth's surface features. The use of 1 arc-second DEMs provides unprecedented detail at the global scale, and their methodology of combining geomorphological ontologies with key derivatives appears to effectively balance noise reduction while preserving important landform characteristics.

However, a notable limitation is the lack of a fully documented methodological scripting procedure (even an example code would be helpful) to enable complete reproducibility of the results. Several Python libraries, such as rasterio, pyjeo, xarray, and numpy, along with GRASS GIS modules, offer matrix filtering procedures and cumulative cost analysis that could facilitate the replication of the methodology in a more transparent way.

The full methodology (AS, TIN, SUI) is novel; however, several issues arise during the processing phase due to the absence of a computational scripting framework that would enhance the rigor of the geocomputation procedure.

**Response:** Thank you for your recognition and comments. In our previous work, we implemented the workflow using ESRI ArcGIS Pro, but due to version differences, some tools may not function consistently across different systems, limiting reproducibility. To address this, we are actively adapting our workflow to open-source alternatives where feasible. We have constructed a Github repository and uploaded a part of tool incorporating Whitebox Geospatial Tools, GRASS GIS, and other open-source software and libraries to enhance accessibility and reproducibility. Due to time constraints, we have not yet provided all the tools, but we will continue to update them in the future. More details can be found in https://github.com/nnu-dta/GBLU-code.For transparency and usability, we have also provided a more detailed explanation about the workflow, including the rationale for constructing the new factor and the detailed calculation processes, in the revised manuscript. (Lines 159-186 and Lines 214-235)

**Below are some geocomputation issues identified in the manuscript:**
**Data pre-processing**
To reduce projection distortion, the authors state:
"Data from latitudes below 70° are transposed onto the Behrmann projection, while data above this threshold are projected onto the Lambert azimuthal equal-area projection."

This approach is reasonable; however, an overlap between the two projection zones is necessary to avoid border effects.

**Response:** Thank you for your reminder. To mitigate border effects, we have implemented an overlapping strategy in our processing. Specifically, we processed the DEMs in 11° × 11° tiles, ensuring that the main 10° × 10° area is used as the final output. This approach helps maintain consistency and minimizes distortions at the transition between projection zones. Related explanation has been added in the revised manuscript. (Lines 136-141)

**Methodology**

Figures 1, 2, and 3 are well designed and effectively illustrate the methods. However, they are not supported by a scripting procedure that can be followed step by step. Additionally, several thresholds (e.g., Tas, Tss) are defined in the methodology but appear to be based on empirical, subjective decisions. It would be preferable to define them using statistical or mathematical criteria.

**Response:** For ease of scripting, we have created a GitHub repository (https://github.com/nnu-dta/GBLU-code) and will continue to update the related tools based on open-source libraries. Additionally, we have supplemented the description of the calculation processes in the revised manuscript. Furthermore, as you mentioned, using statistical or mathematical criteria to define thresholds is an excellent approach. However, given the complex and diverse nature of surface morphology in our study, we attempted histogram-based and mathematical methods but found it challenging to establish a unified standard.

Figures 5–7 are well presented, but it would be beneficial to show the GBLU classification results alongside a transect, similar to Figure 3c, but using real relief data.

**Response:** Thank you for your suggestion. In the revised manuscript, we have optimized the visual appearance of these figures.

Due that the post-processing includes several aggregation/smoothing procedure do you really need to use a 1 arc-second DEM?

**Response:** This is an interesting question. Regarding the use of a 1 arc-second DEM, there is no contradiction between the aggregation procedure and spatial resolution. The aggregation is applied to reduce scattered noise without altering the boundaries generated in our classification. Thus, the final data resolution remains consistent with the original 1 arc-second DEM.

Would be more effective to use 3 arc-second MERIT Hydro in combination with the stream-network Hydrography90m to have a landform classification more in line with existing DEM-derived products?

**Response:** Thank you for your suggestion. While combining 3 arc-second MERIT Hydro with the 90m stream-network Hydrography90m could potentially improve consistency with existing DEM-derived products, the effectiveness remains uncertain due to differences in spatial resolution. Our primary goal is to develop a 1 arc-second landform classification map, and currently, there are no globally available and publicly accessible 30m (1 arc-second) stream network datasets that align with our resolution requirements.

**Projection**

The manuscript states: "Data from latitudes below 71° are transposed onto the Behrmann projection, while data above 69° are projected onto the Lambert azimuthal equal-area projection." However, WGS84 (World Geodetic System 1984) is a geodetic datum and can be represented using either a geographic coordinate system (latitude/longitude, expressed in degrees) or a projected coordinate system (e.g., UTM). The final tif files appear to be stored in the latter, but no specific explanation is provided in the manuscript.

Are the final tif files stored under two separate projections, or have they been homogenized into a single projection? Either approach is valid, but this should be explicitly stated in the manuscript and in the README.txt file available in the Zenodo repository.

**Response:** Thank you for your suggestion. In our processing workflow, we used the Behrmann projection for latitudes below 71° and the Lambert Azimuthal Equal-Area projection for latitudes above 69°. For consistency and ease of use, the final TIFF files have been reprojected into a single coordinate system (EPSG:3857). We have stated this in the manuscript and update the README.md file in the Zenodo repository accordingly.

Additionally, the processing appears to be done in 10° × 10° tiles. What happens at the tile borders? Is there an overlapping procedure in place?

**Response:** As the response above, we have implemented an overlapping strategy in our processing flow. We used DEMs in 11° × 11° tiles, and the main 10° × 10° area is used as the final output. For the boundary, we have manually checked and modified it.

**tif files**

The inclusion of tif file overviews (*.ovr) and a color table palette is appreciated, as they facilitate fast and visually informative rendering. However, it would be useful to include the code legend as metadata within the tif files themselves or at least document it in the README.txt file.

The .aux.xml files store statistical information about the tif files (e.g., mean, median). However, since the tif files contain categorical variables, this statistical information is not particularly useful.

I suggest increasing the grid tile size of the final tif files to 2° × 2° (or even 4° × 4°) to reduce the total number of files. This would simplify tile management, especially for large-scale downloads.

**Response:** Thank you for your reminder. In this version, we have uploaded the README.md file, which now includes explanations of the code meanings and colormap. While the .aux.xml file (which contains statistical information) is not essential for most applications, it is necessary in ArcGIS Pro for rendering data in unique value mode, which enhances usability. Additionally, we have mosaicked the data into 10° × 10° tiles and organized them into folders based on latitude for better accessibility

---

## Author Comment (AC3)

I read the paper "Global basic landform units derived from multi-source digital elevation models at 1 arc-second resolution". There are some interesting aspects, but even if it a technical/data paper there is the need o improvements. Apart from the description of the methodology that is unclear, I think that there are many drawbacks in the paper that require a full restructuring of the work. First, the landforms classification is too simple and in no way reflects the complexity of landscapes. For example, the approach of Iwahashi et al. uses much more information, for example the texture of terrain (even if with a simplified index). The comparison with other methods is debatable both for the different rational behind some methods as well as for the different resolutions. You should at least apply those methods on the same DEMs you used with your approach. Here I suggest some references, to which I refer in the following more detailed comments.

Suggested references

Guth, P.; Kane, M. Slope, Aspect, and Hillshade Algorithms for Non-Square Digital Elevation Models. Transactions in GIS 2021, 25, 2309–2332, doi:10.1111/tgis.12852.

Fisher, P.; Wood, J.; Cheng, T. Where Is Helvellyn? Fuzziness of Multi-Scale Landscape Morphometry. Transactions of the Institute of British Geographers 2004, 29, 106–128.

Trevisani, S.; Guth, P.L. Terrain Analysis According to Multiscale Surface Roughness in the Taklimakan Desert. Land 2024, 13.

Minár, J.; Drăguţ, L.; Evans, I.S.; Feciskanin, R.; Gallay, M.; Jenčo, M.; Popov, A. Physical Geomorphometry for Elementary Land Surface Segmentation and Digital Geomorphological Mapping. Earth-Science Reviews 2024, 248, doi:10.1016/j.earscirev.2023.104631.

Lindsay, J.B.; Newman, D.R.; Francioni, A. Scale-Optimized Surface Roughness for Topographic Analysis. Geosciences (Switzerland) 2019, 9, doi:10.3390/geosciences9070322.

Guth, P.L.; Trevisani, S.; Grohmann, C.H.; Lindsay, J.; Gesch, D.; Hawker, L.; Bielski, C. Ranking of 10 Global One-Arc-Second DEMs Reveals Limitations in Terrain Morphology Representation. Remote Sensing 2024, 16, doi:10.3390/rs16173273.

**Response:**

Thank you for your feedback and suggestions. Following your suggestions, we have reviewed relevant literature and expanded our comparisons to existing landform classification methods and indices, which has significantly enhanced the quality and originality of our paper. Below, we provide a general response to your comments, followed by detailed point-by-point replies

First, regarding the complexity of classification systems, it is important to clarify that our method and the method proposed by Iwahashi emphasize different perspectives. The term "landform" is inherently scale- and context-dependent. For example, "mountain" can represent complete geomorphological entities in general geomorphology or subdivisions emphasizing vertical zonation relevant in climatic and biodiversity research [1]. Iwahashi's classification primarily highlights local variations in terrain features, incorporating a slope level of detail at a smaller scale. This study, however, differs from ours in the classification perspective. We specifically emphasizes force accumulation, mountain ecosystems, and microclimatic gradients before constructing the classification system. GBLU dataset's Level 1 corresponds to the conventional concept of a complete landform entity, while Levels 2 and 3 provide progressively finer-scale morphological information. However, the scale of our finest level remains slightly larger compared to Iwahashi's results. Therefore, while we acknowledge the complexity and effectiveness of the methods used by Iwahashi, our approach differs in terms of the classification perspective and scale, making it suitable for different geomorphological research contexts. Related explanation has been added in the revised manuscript (Lines 99-106).

Secondly, although our approach and that of Iwahashi employ different indicators, the core geomorphological information emphasized in both methods—relief and elevation—is essentially similar. We referred to the excellent work by Iwahashi when constructing the GBLU. Regarding the indicator "texture" mentioned in your comments, Iwahashi defines it as "Texture is calculated by extracting grid cells (here, informally, "pits" and "peaks") that outline the distribution of valleys and ridges in the DEM". We think this indicator differs from the "texture" commonly used in remote sensing studies, such as the gray-level co-occurrence matrix, and is closer to terrain roughness or relief. In our research, we similarly utilized relief but introduced a novel, regional-scale method to measure terrain relief. Furthermore, we did not follow the conventional window-based analysis approach to address scale effects. Instead, we adopted an alternative cumulative perspective for calculating relief, effectively mitigating the scale effects associated with window-based methods. Although it is challenging to precisely determine which approach contains more information, our method captures a similar scope of terrain characteristics as Iwahashi's but through a different analytical strategy. We have added more details about out method and metric in Lines 159-186 and Lines 214-235.

Additionally, it is worth noting that although the segmentation method used by Iwahashi can effectively capture complex terrain characteristics at finer

scales, it involves parameter selection processes that may introduce uncertainties or ambiguities. Similarly, clustering methods can effectively unravel complex relationships among terrain variables, but it has the "black-box" or "gray-box" issues. Specifically, the cluster's results do not inherently possess clear geomorphological meanings, necessitating expert interpretation, as highlighted by Iwahashi and Yamazaki (2022). We greatly appreciate the methods proposed by Iwahashi, but we also recognize that when addressing geomorphological issues, these approaches are not the only feasible solutions.

Regarding method comparisons, we appreciate your comment about using DEMs with differing resolutions. We agree that this issue needs consideration. To address this, we reproduced Iwahashi's classification approach using tools available in SAGA GIS. The results can be found in the following response. We ensured the inclusion of texture metrics emphasized by Iwahashi in our experimental replication. Overall, our results perform better in preserving the integrity of geomorphological features, effectively capturing their macroscopic characteristics and cumulative attributes. The Iwahashi method have good performance in characterizing objects ate smaller scales and but generate relatively fragmented patches in a perspective of the macro scale. In the revised manuscript, these additional analyses and comparisons further clarify our method's robustness and highlight its contributions to broader-scale geomorphological studies.

[1] Evans, I.S., 2012. Geomorphometry and landform mapping: What is a landform?. Geomorphology, 137(1), pp.94-106.

Specific comments (A: author R: reviewer)

A: Lines 67- 69 and also lines 72-74 "Nevertheless, higher DEM data resolution can be regarded as a double-edged sword, in that it at once provides the opportunity for landform mapping at a finer scale while at the same time increasing the challenge of reducing the noise effect (Jasiewicz and Stepinski, 2013) and maintaining the integrity of the identified landforms."
R: I think that the referred problem of noise related to high resolution is a false problem. Apart from the ambiguity of the term "noise" (e.g., noise because of errors in the digital representation, or because you consider noise the fine-scale morphology?), multi-resolution approaches permit to analyze the landscape having control of the "noise" (independently from the interpretation). In addition, surface texture analysis should be an important component of landscape

segmentation approaches (as Iwahashi et al.   or Jasiewicz and Stepinski, 2013) and can be particularly informative when computed at higher resolutions than global DEMs. Apart from the papers you cited I would consider the ones from Fisher Lindsay   and Trevisani

**Response:**

In the original manuscript, our description of the workflow and the factors used was not entirely clear, which may have led to some misunderstandings. In the revised manuscript, we have made the following modifications:

(1) We have clarified and defined the classification objects (Lines 99-106);

(2) We have added explanations for two key factors along with detailed computational steps (Lines 159-186 and Lines 214-235);

(3) We have supplemented the results comparison with additional explanations (Lines 296-308).

We appreciate your valuable comments. We acknowledge that our previous use of the term "noise" might have caused confusion. In fact, we intended to emphasize both data noise (errors) and abrupt terrain changes in our original text, as both significantly affect the classification process and results. To avoid potential misunderstandings, we have revised the original text to "the negative effects of data noise and abrupt terrain variations". (Lines 69-70)

These fine-scale morphological variations have significant value for detailed landform classification, especially at slope or finer scales. The texture employed by Iwahashi is essentially a typical metric emphasize fine-scale morphology which is calculated based on local terrain variability derived from DEMs. However, for geomorphological studies beyond detailed slope-scale analyses, such as vertical mountain zonation, leaving these variations unprocessed would hinder the generation of meaningful classification results. Specifically, the unprocessed fine-scale variability will lead to fragment landscape units and incorrect topographic structures. Therefore, whether such "noise" is beneficial or detrimental depends not solely on data resolution but fundamentally on the specific research context. As we emphasized previously, the GBLU dataset is intended for broader applications in geoscience, particularly in studies focusing on force accumulation, mountain ecosystems, and microclimatic gradients. Under these considerations, appropriate handling or aggregation of these variations becomes necessary.

In practical implementation, the multi-resolution approaches you mentioned indeed provide a feasible solution. By synthesizing terrain characteristics across multiple scales, these approaches can effectively mitigate scale-dependent limitations. However, these methods still inherently face challenges

associated with determining appropriate scales ranges in algorithms. How to select the optimal scale range and properly combine multi-scale terrain features remains a persistent issue. These methods, while widely adopted, are not the only possible solution, and we suggest an alternative approach. Our strategy begins with a step back. Specifically, we consider whether decreasing the reliance on window-based analysis, and then design the novel accumulated slope and relief index.

Regarding texture analysis, we agree that it plays a crucial role in terrain quantification, particularly in multi-scale segmentation approaches. As you noted, a key challenge lies in selecting an appropriate analysis radius or window size. Jasiewicz and Stepinski (2013) also highlighted the difficulty of achieving a universally optimal result using a multi-window approach. After reviewing the terrain texture approach you mentioned, we found that its underlying concept is similar to our relief-related index (previously referred to as the "surface uplift index"). As described by Iwahashi et al (2007), texture is derived by extracting "pits" and "peaks" from a DEM based on elevation differences between the original and a median-filtered DEM. This approach effectively removes high-frequency variations while highlighting terrain features at a local scale. However, it still relies on a predefined window size, which may limit its ability to capture broader topographic patterns. In other words, our methodology and texture-based approaches share a common foundation, as both aim to emphasize topographic relief. Specifically, our approach, which emphasizes regional topographic variations, and texture-based methods, which highlight local terrain variability, represent two complementary strategies aimed at reducing scale-dependent uncertainties in digital terrain analysis.

To more clearly illustrate the differences, we conducted an additional comparative analysis using the Iwahashi classification method implemented in SAGA GIS at the same data foundation (FABDEM) as GBLU. Results based on Iwahashi's method emphasizes local terrain variability, resulting in numerous small-scale geomorphological units. But many of them consist of isolated and fragmented patches—even at the single-pixel scale. While this method effectively captures fine-scale terrain variability, such fragmented landform units pose substantial challenges for macroscopic geomorphological studies, as well as related climate and ecological analyses. Specifically, small and isolated landform units, such as those shown within the highlighted box (black marked in the following figure), cannot support the exploration of macroscale geomorphic patterns due to their limited scale and unclear geomorphological meanings. Additionally, the spatial continuity and

relationships among these fragmented units has been broken and cannot be effectively restored through post-processing techniques, such as filtering.

[Figure]

**Addition response for literature noted by the review:**

Regarding the other papers you mentioned, we have analyzed them as well:

Fisher's study: Similar to the geomorphon approach, it focuses on terrain feature extraction, employing an membership function to resolve classification ambiguities.

Lindsay's study: Introduces a Locally Adaptive Scale-Optimized Surface Roughness Measurement, which applies Gaussian blur to suppress terrain complexity at scales smaller than the filter size.

Trevisani's study: Investigates landform classification in desert regions using multiscale terrain roughness, employing a simplified Multiscale Geostatistical approach to address multiscale effects.

These studies share a common methodology of synthesizing multiscale features by integrating results from multiple window sizes or radius, primarily emphasizing local topographic attributes such as roughness. As we previously discussed, while our approach differs in methodology, it does not conflict with these techniques but rather offers an alternative solution.

Additionally, based on our findings, our dataset demonstrates an improved representation of individual dune features in desert regions compared to Trevisani's approach. While Trevisani's unsupervised classification method provides a more classes, it remains uncertain whether these additional classes

hold strong geomorphological relevance or have meaningful applications in fields such as ecology and environmental studies.

A: Lines 77- 79 "We focus on the classification of basic landforms that emphasizes morphological differences and, in so doing, we present the practical expression of landform ontology at the global scale that offers valuable insights into the Earth'smsurface structure comprising the constellation of landform types and their boundaries."

Lines 80-82. "The objectives of this research are: (1) to construct a global classification system for landforms that integrates geomorphological knowledge, (2) to design a novel framework for global basic landform classification, (3) to develop an automated classification and mapping model for global landforms, and (4) to make available a comprehensive high-resolutiojn dataset of global landform units"

R: I have the feeling that the stated objectives of the research are only partially covered. In regard to 1, I don't see big integration with geomorphological knowledge. In regard to point 3, you are just mapping very simple aggregates of landforms (mountain, hill, plain) that do not represent the complexity of landforms. I think that the work of Iwahashi should be considered the starting point for new approaches, maybe considering additional geomorphometric derivatives. But just working with elevation, even if the algorithm could be interesting, does not seem a step forward and very useful practically. Finally, in regard to (4) I don't think that term "high resolution" can be used with something derived from global DEM at 1 arcsecond resolution.

**Response**:

 After careful consideration, we think that the term "knowledge" could potentially cause misunderstandings. Therefore, in the revised manuscript, we replaced it with "domain consideration of landform-related studies". (Line *)

 To address your comments, we still begin with a discussion of the classification objects. Specifically, the types of landforms used for classification are context-dependent. For example, in subfields of geoscience such as climate and ecological studies, the accumulated effects of energy and materials require a certain continuity of landform objects. This is because accumulated environmental effects typically occur within continuous and coherent units. Additionally, in practical scenarios, an area with slopes slightly steeper than the moderate slope threshold but generally exhibiting gentle trends is not commonly perceived as a "steep slope" by observers. Hence, emphasizing continuity and coherence in landforms aligns better with perceptions and

practical applications as shown in the Figure 1(a). Through this perspective, although "plain", "hill", and "mountain" are commonly used terms, their precise classification at a global scale introduces considerable complexity due to variations in local context and field-dependent definitions. During this process, we need to accept minor local variations to ensure the integrity of geomorphological units. This domain consideration is precisely the original intention behind our earlier emphasis.

[Figure]

Figure 1

Furthermore, it is essential to analyze, from both methodological and result-oriented perspectives, why the classification of "plain hill mountain" poses a complex challenge. Fundamentally, the study can be approached from two scales: the micro-scale and the macro-scale. In geomorphometry, the micro-scale or slope-scale approach emphasizes the capture of detailed terrain variations, as demonstrated in Iwahashi's work. However, a careful examination of Iwahashi's results—whether considering the released dataset or the reproduction of their method on a 30 m DEM (based on your comment)—reveals numerous fragmented geomorphic types, some of which occupy only a single pixel. Even when we synthesize the categories (by converting "gentle"

and "moderate" slopes in Iwahashi's results into "plain" and "steep" and "very steep" slopes into "mountain"), the results still contain a large number of fragmented units (marked by black dot square). From a surveying or terrain measurement standpoint, this may be regarded as an indication of high precision. Nevertheless, for macro-scale landform studies, as well as climate and ecological research related to geomorphology, such fragmented units cannot adequately support the exploration of landform or Earth system patterns. Specifically, these units lack representativeness; analyses based on such units, particularly statistical analyses, are prone to substantial deviations or "outliers" and can significantly impact the performance of subsequent simulation models. More importantly, the structural information of these fragmented patches is difficult to recover (e.g., the connectivity of valley), as indicated by the areas highlighted with red square in the figure.

[Figure]

Figure 2

On this basis, when re-examining "plain hill mountain," what is truly required in our methodology is an increased tolerance for discontinuities or non-typical variations, thereby reducing the occurrence of units with abrupt changes in the results. Consequently, although "plain hill mountain" might sound like a common term, its extraction remains highly complex and need novel method

(Figure 1b-g). Our comparison with the objects and methods in Iwahashi's study is not intended as a competition to determine which approach is more complex; rather, it is aimed at achieving a synergistic enhancement tailored to different research needs.

Finally, we removed the "high resolution" descriptor and revised it to "(4) to make available a comprehensive global dataset of landform units."

A: Lines 91-100
R: The motivations behind the derivation of the simple classification scheme are unclear and someway highly debatable. I don't feel that it is a big deal to just subdivide between mountains, hills and plains. In addition, on the fuzziness of landforms perception and classification I surely would consider the work of Fisher et al.

**Response**:

**Conceptual and perceptive perspective**:

As previously stated, the landform objects in our classification system hold significant importance for ecological and climate research, especially in mountainous regions. More details can also be found in the previous response.

**Technical Perspective**:

In practice, distinguishing between mountains, hills, and plains is not a simple task. For example, plains are not uniformly flat; they can exhibit areas that do not possess typical plain characteristics due to abrupt topographic changes or data errors. When using basic and typical terrain metrics—even with multi-scale approaches—fragmented patches persist, which in turn affect subsequent analyses and the performance of related geographic process simulation models as we noted in the previous response. This phenomenon is particularly pronounced at the interface between mountainous and plain areas (black dot squares in the following figure). Moreover, these fragmented units significantly impact the overall geomorphic structure. While isolated, meaningless pixels can be mitigated through filtering techniques, once structural aspects (such as connectivity) are disrupted, it becomes exceedingly difficult to reconstruct these relationships (red squares in the following figure).

[Figure]

Iwahashi et al. 's method    Our result

Plain
Mountain

We have carefully reviewed Fisher's work you mentioned, which presents an effective method for reducing ambiguity. However, fundamentally, his approach addresses issues arising from scale effects inherent in window-based analyses. As noted earlier, our methodology seeks to "jump out" the window analysis process entirely—in other words, our approach modifies the treatment of ambiguity before any window analysis is performed. We view these as two parallel routes; given the distinct underlying logics, it is challenging to definitively assess which approach is superior. Under our current objectives, we believe our method offers distinct advantages.

A: Lines 107-111 "In this work, the 'Forest and Buildings removed Copernicus DEM' (FABDEM) (Hawker et al., 2022) is the primary data for latitudes 60°S-80°N…"
R: I would be more cautious or at least I would discuss more the selection of FABDEM instead of COPDEM, because some geomorphometric derivatives, are better represented in COP.
See for example Guth et al. In addition, another question is whether structures should be removed in urban landscapes or not.
**Response**:

Thank you for your insightful comment regarding the data selection. We carefully evaluated both FABDEM and COPDEM, considering studies such as Guth et al. and other related work [1]. We found (as also noted by Guth) that FABDEM performed better in digital terrain model (DTM) accuracy tests, which is crucial for accurately classifying natural landforms. In areas with extensive surface cover such as vegetation and buildings, COPDEM's performance is suboptimal. In this study, our goal is to classify natural landforms. Urban landscapes, especially buildings, tend to obscure the natural relief of the terrain. Therefore, we believe it is necessary to select data that have been stripped of building artifacts.

[1] Bielski, C.; López-Vázquez, C.; Grohmann, C.H.; Guth. P.L.; Hawker, L.; Gesch, D.; Trevisani, S.; Herrera-Cruz, V.; Riazanoff, S.; Corseaux, A.; Reuter, H.; Strobl, P., 2024. Novel approach for ranking DEMs: Copernicus DEM improves one arc second open global topography. IEEE Transactions on Geoscience & Remote Sensing. https://doi.org/10.1109/TGRS.2024.3368015

A: Line 117 "knowledge-guided framework…."
R: how? I don't see a relevant integration with expert knowledge.
**Response**:
Thank you for your careful review. As stated earlier, our emphasis is on the considerations for practical applications, particularly the specific needs in climate and ecological studies that are closely related to landforms. Accordingly, we have revised the text to "a new framework", and added additional explanation about the landform objects in our classification in Line 124.

A: Line 119 "calculation of the mountain uplift index (SUI)"
R: I feel that the name "uplift index" is ambiguous, it seems to imply some tectonic uplift. Moreover, see also later comment, it seems a local relief measure.
**Response**:
Thank you for your suggestion. We have renamed it to surface relief index. This metric quantifies the degree of relief, yet it differs significantly from traditional window-based calculations. Instead of evaluating the relative relief within a fixed analysis window, this indicator is designed to measure the relief at any given location across a regional scale. In the revised manuscript, we added detailed explanations of the computational steps. (Lines 214-235)

R: Line 121 What is "factor calculation" ?

**Response:**

We change it to "characteristic quantification".

A: Figure 1, workflow and lines 128-130 "Meanwhile, due to the requirement of calculating landform derivatives, we determine the projection principles as follows: data from latitudes below 70° are transposed onto the Behrmann projection, and the remaining data are transported onto the Lambert azimuth equal-area projection. "

R: To work in a projected system is not a requirement but a choice. In every case if you project DEMs you should discuss all the related intricacies and approximations. See for example Guth and Kane.

**Response**:

We appreciate your comment. While working in a projected system is a choice rather than a requirement, we selected equal-area projections (Lambert Azimuthal Equal Area and Behrmann) to ensure consistency in area-based computations (e.g., using unit area to cartographic synthesis)

As noted by Guth, the Lambert Azimuthal Equal Area projection is well-suited for some regions, as it maintains consistent east-west and north-south spacing when converting arc-second DEMs to projected grids. This minimizes errors in topographic computations, including slope and aspect, which supports our methodological design.

For lower and mid-latitude regions, the Behrmann projection offers reduced scale and shape distortion compared to Lambert projections centered at 45°, providing a better balance between area fidelity and shape preservation. The Lambert projection, however, remains more suitable for mid- and high-latitude regions.

Furthermore, while slope calculation differences between arc-second DEMs and UTM projections are relatively minor (~8-9%), our cumulative slope algorithm accounts for spatial continuity, mitigating potential differences due to DEM projection error.

Finally, to mitigate border effects between the two projection zones, we have implemented an overlapping strategy in our processing. Specifically, we processed the DEMs in 11° × 11° tiles, ensuring that the main 10° × 10° area is used as the final output. This approach helps maintain consistency and minimizes distortions at the transition between projection zones.

R: Figure 2 and related caption. I think it is really difficult to understand how the AS works.

Also the description at lines 149 -160 is unclear to me: "The AS is calculated as the minimum cumulative cost of each position to the nearest landform core along a specific path…"

How is computed cost? The cost of doing what? I don't see how geomorphological knowledge enters in the method, it seems an heuristic approach.

**Response:**

In the revised manuscript, we have revised Figure 2 and the corresponding text with a detailed explanation of why we use cost for AS calculation and how it is calculated. The specific explanations can be found in Lines 159-186.

A: Lines 176-178 "However, commonly employed indices reflecting topographic relief are achieve using a window of fixed size such as 3×3, 5×5 pixels, or larger (Maxwell and Shobe, 2022), a method that fails to account for geomorphological semantics, and which therefore disregards the integrity of a mountain. Window size has a significant impact on results of relief calculation."

R: but adopting multiscale approaches this issue can be resolved.

**Response:**

Multiscale approaches are an effective method that can mitigate multi-scale effects and integrate features across different scales. However, it is difficult to assert that they can fully resolve the issue. Regarding the fundamental differences between our approach and multiscale analysis, we have already addressed this in our previous responses. In summary, our method computes global features over an entire region, whereas multiscale approaches integrate local features at various scales. Although both methods aim toward similar objectives, the classification targets in our study differ from those in research that emphasizes local features. Multiscale approaches are difficult to completely resolve the issues we have identified.

Based on your suggestions, we have supplemented our manuscript with an experiment in which we reproduced landform classification using Iwahashi's tool published on SAGA with FABDEM data. The results, as shown in the figure below, indicate that even with multiscale approaches, the final outputs still exhibit a substantial number of fragmented units. We think that while such results may be advantageous for representing landform objects at the slope scale, they could have negative implications when classifying landform objects at a relatively macro scale. For macro-scale landform studies, as well as climate and ecological research related to geomorphology, such fragmented units (marked by black square in the following figure) cannot adequately support the exploration of landform or Earth system patterns. Specifically, these units lack

representativeness; analyses based on such units, particularly statistical analyses, are prone to substantial deviations or "outliers" and can significantly impact the performance of subsequent simulation models. Moreover, these fragmented units significantly impact the overall geomorphic structure. While isolated, meaningless pixels can be mitigated through filtering techniques, once structural aspects (such as connectivity) are disrupted, it becomes exceedingly difficult to reconstruct these relationships.

[Figure]

A: Line 183 "In quantitative analysis, it is crucial to consider the underlying terrain of mountains to accurately assess changes in elevation."

R: unclear.

**Response:**

We revised this sentence to "Therefore, we propose a new method for relief quantification method which do not rely on the traditional window-based calculation. In this paper, the surface relief index proposed in this paper is defined as the degree of relative relief to the flat areas surrounding the mountain. We regard the elevation at the foot of the mountain as the base elevation and then calculate the elevation difference between each position on the mountains and the base elevation". (Line 220)

A:Lines 185 "surface uplift index (SUI)"

R: your index seems a local relief index on which there is a huge literature (see for example Minar and cited reference therein…).

**Response:**

This metric quantifies the degree of relief, yet it differs significantly from traditional window-based calculations. Instead of evaluating the relative relief within a fixed analysis window, this indicator is designed to measure the relief at any given location across a regional scale. In the revised manuscript, we revised it to "we propose a new method for relief quantification method which do not rely on the traditional window-based calculation. In this paper, the relief is defined as the degree of relative relief to the flat areas surrounding the mountain. We regard the elevation at the foot of the mountain as the base elevation and then calculate the elevation difference between each position on the mountains and the base elevation. Compared to the traditional method of relief calculation (e.g., difference in elevation within a particular window size), the surface relief index proposed in this paper considers the vertical elevation differences between the surface and the mountain base, which is more suitable for the objectives in landform-related studies such as mountainous climate and biodiversity". (Lines 217-222)

A:Lines 188-189 "SUI considers the vertical elevation differences between the surface and the mountain base, which is more consistent with the human perception of mountain morphology."

R: The human perception is multiscale, so it just depends from the target of the analysis.

**Response:**

Thank you for your comment. We changed it to "the surface relief index considers the vertical elevation differences between the surface and the mountain base, which is more suitable for the objectives in landform-related studies such as mountainous climate and biodiversity". (Lines 221-222)

R: Lines 190-203. Not able to follow.

**Response:**

Thank you for your suggestion. We have revised this section, adjusting the logic and incorporating detailed computational steps. (Lines 223-237)

A: Lines 241-242 "Figure 4 shows the global landform classification results based on the abovementioned framework. This hierarchical dataset provides a

more comprehensive understanding of the Earth surface"

R: A more comprehensive with respect to which method? Or with respect to which reference dataset? Honestly the earth's surface is a little bit more complex. Apart from the issues with deserts you mention, for instance big depressed areas or volcanic environments are not represented.

**Response:**

Regarding the specific advantages of GBLU, we have provided a more detailed explanation (Lines 296-308). (1) GBLU demonstrates exceptionally complete valley results with more accurate boundary and shape delineation. Valleys are critical landforms in geomorphology and related ecological studies, and they represent a category of depression-type features. As mentioned earlier, the other methods tend to emphasize accuracy at the slope scale, but for features that require a higher classification level with an emphasis on completeness and boundary accuracy, GBLU performs better. (2) as you pointed out, in volcanic regions, GBLU does not display certain erosional signatures that are apparent in Iwahashi's results. Our approach captures more transitional phenomena between volcanic areas and the surrounding terrain. These revisions and the related descriptions have been incorporated into the manuscript.

A:Lines 244-245 "The selected regions contain examples of the main landforms on Earth, as well as transition areas of different landforms."

R: Yes, in the selected regions there are interesting patterns, but your approach does not characterize/distinguish these.

**Response:**

Thank you for your suggestion. Regarding the results shown in Figure 5, we have added additional explanations in the manuscript. In mountainous regions, GBLU presents a more complete depiction of valleys and peaks, which together form the fundamental structure for expressing mountain. Meanwhile, in desert areas, GBLU clearly reveals the distribution of dunes and interdune regions. Based on these results, we can currently provide a foundational outcome that supports visual differentiation. However, if the focus is on quantitative indicators, unfortunately we have not introduced a metric for quantifying landform patterns in this paper. We think that such an analysis may extend beyond the core scope of the current work, and we plan to conduct more in-depth analyses of landform patterns in future studies.

A: "The abundant textural information provided by GBLU"

R: I don't see how your approach contains textural information in the sense of Iwahashi or Trevisani.

**Response:**

What we intend to convey here is the basic textural information of landform composition as observed visually, rather than a specific metric as used by Iwahashi or Trevisani. To avoid any misunderstanding, we have revised the description to "the information on the landform composition".

A: 259 "significant improvement achieved by applying GBLU is the increased detail in representing terrain features."

R: I see a very simple representation of landforms, but any indicator of patterns/texture is totally missing.

**Response:**

Thank you for your suggestion. Regarding our rationale for selecting these research objects and comparing our method with texture-based method, we have provided detailed explanations in our previous responses—please refer to those for specific details. In the manuscript, we have also supplemented the discussion with additional explanations, including the complexity of landform objects and detailed steps for terrain factor calculations.

R:

Section 3.2

This section has a lot of issues. You need to describe reference data (refdata) in the text not in the captions. Most importantly, it does not make too much sense to compare classifications performed at different resolutions or with different DEMs, given the different generalization levels of the landscape. Regarding Iwahashi you could apply the method to the same data you used in the analysis (if I'm not wrong it is implemented in SAGA). In addition, the method of Iwahashi et has been designed to take into account different aspects of morphology, including texture. It is not just based on elevation and slope.

**Response:**

Thank you for your suggestion. First, in the revised manuscript, we have replaced "refdata" with specific citations.

Second,we have reproduced the classification results using Iwahashi's method at 30m resolution based on the tool in SAGA, ensuring a more direct and meaningful comparison. The results, as shown in the figure below, indicate that even with Iwahashi's approach, the final outputs still exhibit a substantial

number of fragmented units. We think that while such results may be advantageous for representing landform objects at the slope scale, they could have negative implications when classifying landform objects at a relatively macro scale. For macro-scale landform studies, as well as climate and ecological research related to geomorphology, such fragmented units (marked by black square in the following figure) cannot adequately support the exploration of landform or Earth system patterns. Specifically, these units lack representativeness; analyses based on such units, particularly statistical analyses, are prone to substantial deviations or "outliers" and can significantly impact the performance of subsequent simulation models. Moreover, these fragmented units significantly impact the overall geomorphic structure. While isolated, meaningless pixels can be mitigated through filtering techniques, once structural aspects (such as connectivity) are disrupted, it becomes exceedingly difficult to reconstruct these relationships.

Additionally, it is worth noting that although the segmentation method used by Iwahashi can effectively capture complex terrain characteristics at finer scales, it involves parameter selection processes that may introduce uncertainties or ambiguities. Similarly, clustering methods can effectively unravel complex relationships among terrain variables, but it have the "black-box" or "gray-box" issues. Specifically, the cluster's results do not inherently possess clear geomorphological meanings, necessitating expert interpretation, as highlighted by Iwahashi and Yamazaki (2022). We greatly appreciate the methods proposed by Iwahashi, but we also recognize that when addressing geomorphological issues, these approaches are not the only feasible solutions.

[Figure]

**Ours _(GBLU)_ L3 result (30m)**    **Iwahashi et al.'s method (30m)**

- _GBLU Level 3 (30m)_

| | | |
|---|---|---|
| Low-altitude plain | Low-altitude low-relief mountain | Low-altitude high-relief mountain |
| Middle-altitude plain | Middle-altitude low-relief mountain | Middle-altitude high-relief mountain |
| High-altitude plain | High-altitude low-relief mountain | High-altitude high-relief mountain |
| Highest-altitude plain | Highest-altitude low-relief mountain | Highest-altitude high-relief mountain |
| Low-altitude hill | Low-altitude middle-relief mountain | Middle-altitude highest-relief mountain |
| Middle-altitude hill | Middle-altitude middle-relief mountain | High-altitude highest-relief mountain |
| High-altitude hill | High-altitude middle-relief mountain | Highest-altitude highest-relief mountain |
| Highest-altitude hill | Highest-altitude middle-relief mountain | |

- _Iwahashi et al.'s (30m)_

| | |
|---|---|
| Gentle slope, coarse texture, low convexity | Steep slope, coarse texture, low convexity |
| Gentle slope, fine texture, low convexity | Steep slope, fine texture, low convexity |
| Gentle slope, coarse texture, high convexity | Steep slope, coarse texture, high convexity |
| Gentle slope, fine texture, high convexity | Steep slope, fine texture, high convexity |
| Moderate slope, coarse texture, low convexity | Very steep slope, coarse texture, low convexity |
| Moderate slope, fine texture, low convexity | Very steep slope, fine texture, low convexity |
| Moderate slope, coarse texture, high convexity | Very steep slope, coarse texture, high convexity |
| Moderate slope, fine texture, high convexity | Very steep slope, fine texture, high convexity |

Finally, regarding texture, we have provided detailed explanations in our previous responses. In our method, while elevation and slope serve as the foundational parameters, we have introduced innovative elements—particularly new indices—that extend the range of information used beyond just these basic variables. While we acknowledge that it is an excellent metric, it does not represent the only viable solution. For further details, please refer to our earlier responses.

---

## Referee Report (RR1)

My original review was quite favourable because of the innovative way to delimit plains and mountains and how the problem of the size of moving windows (NAW) in neighbourhood analysis had been tackled. I commented that the calculated units did not represent landforms in my opinion, but rather slope and elevation classes within the three broad landform categories: plains, hills and mountains. This point of view is not shared by the authors, but for me it is fundamental as I show in the figure below.

The problem of mountain valleys classified as low hills persists, and some 'landform' categories regroup fundamentally different (convex -  such as ridges - and concave - such as valleys -) topographical shapes. The morphological aspect of the 'landforms' is therefore not adequately captured in this classification, which basically represents units separating plains from hills (problematic, see below) and mountains, combined with slope and elevation classes. By focusing on the high precision of the result through the use of high resolution DTM, the product is of high precision of the mapping units but low accuracy of their classification in terms of landform. It would have been a good thing to validate the proposed landform classification with a number of experts, for example by extracting classified GBLU terrain units of a representative sample of small areas (e.g. 50 * 50 km, like in the figure shown below)  and asking whether the proposed units resonated with their understanding of the landscape.

While acknowledging that the macro delimitation between plains and mountains (L1) is useful and based on an original approach, I doubt, as claimed, that anyone wishing to study erosion, microclimate or ecological zones locally or regionally would use the GBLU for these purposes, but rather stick to a DTM and derived measures such as ruggedness, slope, orientation etc.

The proposed solutions to the concerns I advanced in my first reviews are purely cosmetic and do not address them fundamentally. I don't think for example that renaming 'mountain' to 'mountain slope' addresses the fact that the mapped units do not represent landforms. If you take a DTM and colour the elevation zones, the patterns you get are almost identical to the units of the GBLU. I don't see a clear advantage of using the GBLU over DTMs and elevation zones.

I am a bit challenged by the fact that the authors present a global landform map, but are fine with classifying a valley(floor) as a hill and then leave it up to the discretion of the users to reclassify their product to identify the actual landforms (such as valleys) they are interested in. If you look at the product and its classifications in a mountain region such as the Alps, the sequence of 'landforms' as one goes from low (valley bottom) to high (mountain peak) is: 1.0.1. (which does not figure in the legend, but I suppose it is 1.1.1.) low altitude plain (which would actually correspond to valley floor), 2.1.1., low altitude hill (which would actually correspond to valley or lower foot slope, depending on the width of the valley), 2.2.1. low altitude low relief mountain slope, 2.2.2. middle altitude low relief mountain slope, 2.3.2., 2.4.2., 2.5.2. and 2.5.3. These last classes mainly represent different elevation and slope classes on the slopes of a large mountain.

[Figure]

This figure from the Alps illustrates what is problematic from my point of view with the 'landform' classification proposed in the GBLU. The landform class 2.2.2. 'Middle-altitude low-relief mountain slope', highlighted in purple, identifies lower slopes or valley bottoms (concave shapes) in the Alps while it identifies ridges (convex shapes) in the alpine foreland. Then, there is an intermediate case where 2.2.2. seems to correspond to mid-slopes of ridges. Ontologically, morphologically, visually and genetically these landforms have very little in common, apart from a certain elevation and slope class. Basically, depending on where you look on the map class 2.2.2. is either a high mountain valley floor (1), a high mountain valley slope (2), a high mountain foot slope (3), a low mountain crest or ridge (4), or a low mountain slope (5).

I still like the innovations introduced to map plains and rugged terrain in great detail but I object to the presenting product's map units as representing landforms.

---

## Author Response (AR2)

**Editor**

Dear Authors,

On previous reviewer and one new reviewer have now considered your revised manuscript. Whilst the new reviewer has a generally favourable view of the manuscript, the original reviewer has more substantial concerns.

I have therefore decided to reconsider the manuscript following major revisions. In addition to responding to the useful comments and suggestions made, I would be grateful if you could carefully consider the claim that the products presented are truly "landform maps", which both reviewers from this round call into question.

Best wishes,

James Thornton

**Response:**

Thank you very much for your review and support. In response to your and reviewers' comments, we have carefully addressed each point and would like to provide further clarification on the issue you raised regarding whether our results can be considered a landform map.

The primary goal of this study is to characterize the morphological characteristics of the land surface, particularly the vertical variation and relief intensity across different landforms, which are a key component of landform classification. However, we fully acknowledge that in a stricter geomorphological sense, the term "landform" often refers to specific and named units such as valleys or plateaus, as highlighted by Reviewer #1. After carefully consideration and reviewing of related literatures, **we recognize that our understanding of the terminology was imprecise. In light of the comments and guidance from relevant literature [1,2], we have revised the terminology throughout the manuscript, referring to our classification results as relief classes instead of landforms.** The concept of relief classes emphasizes a morphological representation of the terrain, focusing on the vertical variation and intensity of surface undulation rather than specific indices. These relief features play essential roles in regulating energy flow and material transport across the Earth's surface and have important implications for geomorphic processes, hydrological balance, and human activities [3-5]. **In this context, we divided the global land surface into two major types: flat terrain and rugged terrain, which approximately correspond to the lowland and mountain categories described in [1,6].**

Moreover, to address the concern raised about the limitations of altitude banding, we revised the classification system. **In the updated version, we merged the previous second- and third-level categories within rugged regions, shifting the emphasis toward relative relief, in line with the reviewers' suggestions.** For flat regions, we retained elevation-based

classification as recommended in previous studies, to better capture subtle variations within plains. We also updated the publicly available datasets to align with the revised classification system. **Additionally, in the data usage section, we have added a comparative analysis with runoff data [7], a key environmental factor related to surface hydrology, to demonstrate the applicability of the relief classes in ecological and hydrological studies.**

Furthermore, we have carefully revised the issues related to textual descriptions and figures throughout the manuscript. Corresponding updates have also been made to the related files in the data repository to ensure consistency with the revised content.

We sincerely hope that these revisions improve the rationality and clarity of the manuscript. Please refer to the detailed point-by-point responses for more information. Thank you very much for your thoughtful guidance and support.

[1] Viviroli, D., Kummu, M., Meybeck, M., Kallio, M., & Wada, Y. (2020). Increasing dependence of lowland populations on mountain water resources. Nature Sustainability, 3(11), 917-928.

[2] Meybeck, M., Green, P., & Vörösmarty, C. (2001). A new typology for mountains and other relief classes. Mountain research and development, 21(1), 34-45.

[3] Thornton, J. M., Snethlage, M. A., Sayre, R., Urbach, D. R., Viviroli, D., Ehrlich, D., Muccione, V., Wester, P., Insarov, G., & Adler, C. (2022). Human populations in the world's mountains: Spatio-temporal patterns and potential controls. PLoS One, 17(7), e0271466.

[4] Zhou, Z., & Chen, Y. (2025). How urban land expansion alters terrain in mountainous and hilly areas: An empirical study in China. Geography and Sustainability, 100304.

[5] Chen, Y., Yang, X., Fu, H., Li, C., & Tang, G. (2025). Characterizing spatial patterns and regionalization of anthropogenic landforms using multi-source geospatial data: Insights from Loess Plateau of China. Geomorphology, 478, 109708.

[6] Viviroli, D., Archer, D. R., Buytaert, W., Fowler, H. J., Greenwood, G. B., Hamlet, A. F., Huang, Y., Koboltschnig, G., Litaor, M.I., López-Moreno, J.I., & Woods, R. (2011). Climate change and mountain water resources: overview and recommendations for research, management and policy. Hydrology and Earth System Sciences, 15(2), 471-504.

[7] Sujud, L. H., & Jaafar, H. H. (2022). A global dynamic runoff application and dataset based on the assimilation of GPM, SMAP, and GCN250 curve number datasets. Scientific Data, 9(1), 706.

**Reviewer #1**

**The comment in the email:**
My original review was quite favourable because of the innovative way to delimit plains and mountains and how the problem of size of moving windows (NAW) on had been tackled. I commented that the calculated units did not represent landforms in my opinion, but rather slope and elevation classes. This point of view is not shared by the authors, but for me it is essential as I show in the report. The problem of mountain valleys classified as low hills persists, and some 'landform' categories regroup fundamentally different (convex - such as ridges - and concave - such as valleys -) shapes. The form aspect of the landforms is therefore not adequately captured in this classification, which basically represents units combining slope and elevation classes.

**Comments in the attached PDF:**
My original review was quite favourable because of the innovative way to delimit plains and mountains and how the problem of the size of moving windows (NAW) in neighbourhood analysis had been tackled. I commented that the calculated units did not represent landforms in my opinion, but rather slope and elevation classes within the three broad landform categories: plains, hills and mountains. This point of view is not shared by the authors, but for me it is fundamental as I show in the figure below.

The problem of mountain valleys classified as low hills persists, and some 'landform' categories regroup fundamentally different (convex - such as ridges - and concave - such as valleys -) topographical shapes. The morphological aspect of the 'landforms' is therefore not adequately captured in this classification, which basically represents units separating plains from hills (problematic, see below) and mountains, combined with slope and elevation classes. By focusing on the high precision of the result through the use of high resolution DTM, the product is of high precision of the mapping units but low accuracy of their classification in terms of landform. It would have been a good thing to validate the proposed landform classification with a number of experts, for example by extracting classified GBLU terrain units of a representative sample of small areas (e.g. 50 * 50 km, like in the figure shown below) and asking whether the proposed units resonated with their understanding of the landscape.

While acknowledging that the macro delimitation between plains and mountains (L1) is useful and based on an original approach, I doubt, as claimed, that anyone wishing to study erosion, microclimate or ecological zones locally or regionally would use the GBLU for these purposes, but rather stick to a DTM and derived measures such as ruggedness, slope, orientation etc. The proposed solutions to the concerns I advanced in my first reviews are purely cosmetic and do not address them fundamentally. I don't think for example that renaming 'mountain' to 'mountain slope' addresses the fact that the mapped units do not represent landforms. If you take a DTM and colour the elevation zones, the patterns you get are almost identical to the units of the GBLU. I don't see a clear advantage of using the GBLU over DTMs and elevation zones.

I am a bit challenged by the fact that the authors present a global landform map, but are fine with classifying a valley(floor) as a hill and then leave it up to the discretion of the users to reclassify their product to identify the actual landforms (such as valleys) they are interested in. If you look at the product and its classifications in a mountain region such as the Alps, the sequence of 'landforms' as one goes from low (valley bottom) to high (mountain peak) is: 1.0.1. (which does not figure in the legend, but I suppose it is 1.1.1.) low altitude plain (which would actually correspond to valley floor), 2.1.1., low altitude hill (which would actually correspond to valley or lower foot slope, depending on the width of the valley), 2.2.1. low altitude low relief mountain slope, 2.2.2. middle altitude low relief mountain slope, 2.3.2., 2.4.2., 2.5.2. and 2.5.3. These last classes mainly represent different elevation and slope classes on the slopes of a large mountain.

[Figure]

This figure from the Alps illustrates what is problematic from my point of view with the 'landform' classification proposed in the GBLU. The landform class 2.2.2. 'Middle-altitude low relief mountain slope', highlighted in purple, identifies lower slopes or valley bottoms (concave shapes) in the Alps while it identifies ridges (convex shapes) in the alpine foreland. Then, there is an intermediate case where 2.2.2. seems to correspond to mid-slopes of ridges.

Ontologically, morphologically, visually and genetically these landforms have very little in common, apart from a certain elevation and slope class. Basically, depending on where you look on the map class 2.2.2. is either a high mountain valley floor (1), a high mountain valley slope (2), a high mountain foot slope (3), a low mountain crest or ridge (4), or a low mountain slope (5).

I still like the innovations introduced to map plains and rugged terrain in great detail but I object to the presenting product's map units as representing landforms.

**Response:**
   We sincerely apologize for not fully understanding your concern in the first-round revision. After receiving your valuable feedback, we conducted a thorough review of the relevant literature and carefully examined how different studies define and classify landforms or related geomorphic objects. In this revision, we made substantial adjustments to both the classification

targets and the classification system.

First, we revised the terminology used to describe our classification targets. **In this study, our primary focus is on capturing morphological characteristics of the land surface, particularly the vertical variation and relief intensity across different landforms.** These features are highly relevant to ecological studies involving runoff and watershed dynamics [1,2], where similar units are often referred to as landforms (for example, the Global Landform Classification dataset provided by the European Soil Data Centre [3], which has the similar targets with our study). However, as you pointed out, these terms may not fully align with a strict geomorphological definition. **To address this issue, and following guidance from relevant literature, we have revised the terminology in our manuscript to refer to the results as *relief classes* rather than landforms.** This shift could reflect the nature of our classification while avoiding conceptual ambiguity. To further enhance clarity and practical value, we have added illustrations in the data usage section to demonstrate the applicability of our results. **We have expanded our discussion of how the derived relief classes can support ecological studies—particularly those focused on mountain-lowland runoff balance and water resource allocation.**

**Second, we adjusted our classification terminology and system by referencing the Global Landform Classification dataset provided by the European Soil Data Centre (ESDAC)**, particularly the terminologies and system proposed by [3]. These changes were made in alignment with your suggestions and to ensure consistency with established international classification schemes.

Please refer to the detailed responses and revised manuscript for further clarification.

**(1) Regarding the Classification Targets**

In this study, our primary focus is on capturing morphological characteristics of the land surface, particularly the vertical variation and relief intensity across different landforms. However, as you pointed out, some of the units in our classification results may diverge from strict geomorphological terms in landform taxonomy. **Therefore, in light of your feedback and the further review of relevant literature [1-3], we acknowledge that the outcomes of our study are more appropriately aligned with the typology of relief classes.** Relief, as a fundamental component in shaping Earth surface processes, provides a more suitable conceptual basis for interpreting our classification results. The relief classes are significant in playing a critical role in regulating energy flows and material transport across terrestrial environments and exerting significant influence on geomorphic evolution, hydrological balance, and human activity [1,4,5]. The concept of relief emphasizes a terrain representation perspective, focusing on the overall morphological variation of the surface rather than referring to a specific relief index. In our study, AS (accumulated slope) is used to characterize surface roughness, while SRI (slope relief index) quantifies terrain undulation. Through methodological innovations in the calculation of both indices, we have achieved a more refined and accurate representation of relief features. This refinement forms the basis for our classification approach.

**To support this interpretation, we have incorporated a comparative analysis between our classification results and global runoff data [6]**. The comparison reveals a strong spatial consistency, especially in mountainous regions. As demonstrated in the newly added figures (Figure 1 in this file), both our classification and the runoff data exhibit similar spatial gradients

and distribution patterns. We have added related explanation in the data usage section of the revised manuscript. Importantly, runoff reflects one of the key hydrological components of the Earth system. By providing more detailed and spatially refined relief classes, our study contributes knowledge that can help advance research in Earth system science, particularly where accurate terrain-based ecological zoning is critical. Given the relatively coarse spatial resolution (approximately 250 m) of existing global runoff datasets, we believe our classification could serve as a valuable base for runoff downscaling and related ecological modeling. Moreover, our classification results contain the ability to support ecological studies focused on hydrological supply-demand imbalances across mountainous and lowland regions such as the issue discussed in [1,7].

[Figure]

**Figure 1**. The spatial distribution of the global relief classes (GRC) and surface runoff in different areas. a and b show the GRC L2 for the Rocky Mountains in North America, while c and d display the corresponding runoff patterns in the same region. e and f show the GRC L2 for the Andes Mountains in South America, while g and h display the corresponding runoff patterns in the same region. i and j show the GRC L2 and runoff in the southern areas of the Himalayas, while k and l display the corresponding runoff patterns in the same region.

**(2) Regarding the Classification System**

In response to your valuable suggestion that traditional geomorphological terms such as plain, hill, and mountain may not precisely represent our derived units, we have revised the terminology to better reflect the actual properties of the classified surfaces. **We now adopt the terms "flat terrain" and "rugged terrain" to denote relatively smooth versus rough**

**terrain, respectively. On this basis, we further subdivided the surface into multiple relief classes, as detailed in the manuscript.** This revised terminology is more commonly used in ecological and climate-related research, where terrain relief rather than strict landform taxonomy is emphasized. We hope this terminology adjustment can clarify the nature of our classification results.

In response to your comments, as well as feedback from the second reviewer and insights from related literature, we have revised the classification scheme accordingly. **First, based on slope and AS, we divided the land surface into two major types: flat terrain and rugged terrain** (corresponding to the mountain and lowland categories described in [1]. Meanwhile, as you noted, introducing elevation-based classification—especially mountainous or rugged regions—can result in ambiguous or difficult-to-interpret units in the context of geomorphological or terrain-based research. Therefore, **in the revised manuscript, we merged the level 2 and 3 within rugged areas.** The updated classification in these regions now emphasizes relative relief, also aligning with the suggestion from Reviewer #2. **In flat regions, we retained elevation-based classification, as recommended by previous studies, to further differentiate plains.** As a result, the revised classification framework now includes two hierarchical levels, capturing the relief characteristics of the land surface in a more interpretable and practical manner.

Following these adjustments, we also updated the released classification dataset, and the associated data repository has been updated accordingly (https://doi.org/10.5281/zenodo.15641257).

**Table 1**. Classification of global relief.

| L1 | Code1 | Colors (RGB) | | L2 | Code2 | Colors (RGB) | | Note |
|----|-------|--------------|---|-----|-------|--------------|---|------|
| Flat terrain | 1 | 129,168,0 | | Low-altitude flat terrain | 11 | 90,138,55 | | Classifying L2 flat lad based on the altitude. |
| | | | | Middle-altitude flat terrain | 12 | 209,235,152 | | |
| | | | | High-altitude flat terrain | 13 | 237,242,179 | | |
| | | | | Highest-altitude flat terrain | 14 | 213,217,164 | | |
| Rugged terrain | 2 | 255,255,190 | | Low-relief rugged terrain | 21 | 230,216,106 | | Classifying L2 rugged lad based on the surface relief index. |
| | | | | Gentle-relief rugged terrain | 22 | 244,100,18 | | |
| | | | | Moderate-relief rugged terrain | 23 | 220,0,0 | | |
| | | | | High-relief rugged terrain | 24 | 86,20,24 | | |
| | | | | Very high-relief rugged terrain | 25 | 255,255,255 | | |

**(3) Regarding the Use of Classification Results and Quantitative Metrics**

We acknowledge that quantitative metrics describing land surface morphology—such as slope, relief index, and curvature—are widely used across geomorphological and environmental studies. **However, in non-geomorphology domains,researchers may lack the technical background to interpret or transform continuous terrain indices into meaningful patterns.** Providing structured, interpretable relief classifications facilitates broader integration and discovery. In addition, directly using topographic indices in non-geomorphology domains may limit effective knowledge discovery as we also mentioned in the main text. For example, runoff estimation often relies on slope-based calculations, which can yield highly fragmented spatial outputs. In contrast, **relief classes offer a generalized representation that aligns better with ecological or hydrological zoning, enhancing interpretability and usability in downstream applications.** Therefore, we think that the availability of derived categorical data (i.e., relief classes) is also important, particularly for applications involving large-scale pattern recognition or interdisciplinary studies such as biodiversity, water sources, population distribution, or urban development [5,8,9].

[1] Viviroli, D., Kummu, M., Meybeck, M., Kallio, M., & Wada, Y. (2020). Increasing dependence of lowland populations on mountain water resources. Nature Sustainability, 3(11), 917-928.

[2] Viviroli, D., & Weingartner, R. (2004). The hydrological significance of mountains: from regional to global scale. Hydrology and earth system sciences, 8(6), 1017-1030.

[3] Meybeck, M., Green, P., & Vörösmarty, C. (2001). A new typology for mountains and other relief classes. Mountain research and development, 21(1), 34-45.

[4] Thornton, J. M., Snethlage, M. A., Sayre, R., Urbach, D. R., Viviroli, D., Ehrlich, D., Muccione, V., Wester, P., Insarov, G., & Adler, C. (2022). Human populations in the world's mountains: Spatio-temporal patterns and potential controls. PLoS One, 17(7), e0271466.

[5] Zhou, Z., & Chen, Y. (2025). How urban land expansion alters terrain in mountainous and hilly areas: An empirical study in China. Geography and Sustainability, 100304.

[6] Sujud, L. H., & Jaafar, H. H. (2022). A global dynamic runoff application and dataset based on the assimilation of GPM, SMAP, and GCN250 curve number datasets. Scientific Data, 9(1), 706.

[7] Viviroli, D., Archer, D. R., Buytaert, W., Fowler, H. J., Greenwood, G. B., Hamlet, A. F., Huang, Y., Koboltschnig, G., Litaor, M.I., López-Moreno, J.I., & Woods, R. (2011). Climate change and mountain water resources: overview and recommendations for research, management and policy. Hydrology and Earth System Sciences, 15(2), 471-504.

[8] Nogués-Bravo, D., Araújo, M. B., Errea, M. P., & Martínez-Rica, J. P. (2007). Exposure of global mountain systems to climate warming during the 21st Century. Global environmental change, 17(3-4), 420-428.

[9] Chen, Y., Yang, X., Fu, H., Li, C., & Tang, G. (2025). Characterizing spatial patterns and regionalization of anthropogenic landforms using multi-source geospatial data: Insights from Loess Plateau of China. Geomorphology, 478, 109708.

**Reviewer #2**

Review by Dr. Ian S. Evans of Durham University, of ESSD-2024-401
Global basic landform units derived from multi-source digital elevation models at 1 arc-second resolution, by Yang et al.

With a major computing effort, ~90 m DEMs are processed globally (82ºN to 88ºS) to outline landform areas at three levels (2, 6 and 23 classes). Initially a dichotomy is mapped: plain cores with slopes of below 1.5º to 3º are extended along traverses until 'accumulated slope' exceeds a threshold. This defines the division between plains and mountains. At a second level, 'mountains' are divided into 5 surface relief classes, of which the lowest is redefined as 'hills'. In level 3, these 6 classes are subdivided into 23 on the basis of 4 elevation or relief bands. (Perhaps some of this more precise description could be included in the abstract.)

**Response:** Thank you for your valuable comments. We have revised the abstract to include more specific details as suggested. We would also like to clarify that the base DEM used in this study is the 1 arc-second resolution (~30 m), not the 90 m resolution as mentioned.

The global results in Fig. 4 produce familiar patterns, in which level 2 essentially maps relief. The only strange aspect is that both ice sheets are mapped as plains, with the narrowest of 'hill' fringes. The complexity of the level 3 map is quite rightly explored in detailed extracts in Figs. 5 – 8. In Fig. 5, c successfully distinguishes hills from plains, but some boundaries in 5d, e, f and g (and Fig. 6) resemble elevation contours rather than landform boundaries. In Fig. 7 there are huge differences between K1, K2 and K3 for the Andes, that are difficult to understand: K3 seems to produce reasonable results for the Alps and Himalayas. In Fig. 8c the area mapped as 'hills' is a thin rim around the mountains – a 'mountain fringe' rather than hills.

**Response:**

Thank you for your insightful suggestion. **Another reviewer raised a similar concern, noting that our classification targets may differ from strictly defined geomorphological landform units.** In this study, our primary focus is on capturing morphological characteristics of the land surface, particularly the vertical variation and relief intensity across different landforms. These features are highly relevant to ecological studies involving runoff and watershed dynamics [1], where similar units are often referred to as landforms. **After carefully considering the reviewers' comments and conducting a further review of relevant literature, we acknowledge that the outcomes of our study are more appropriately aligned with the typology of *relief classes* [1,2]**. Relief, as a fundamental component in shaping Earth surface processes, provides a more suitable conceptual basis for interpreting our classification results. his shift more accurately reflects the nature of our classification while avoiding conceptual ambiguity. Meanwhile, to further enhance clarity and practical value, we have added illustrations in the data usage section to demonstrate the applicability of our results.

In addition, following the adjustment of classification targets, we have also addressed the concern you raised regarding the use of the term "hill" and "mountain," which may not precisely represent the derived units. Drawing on previous literature [2], we have revised the terminology to better reflect the actual properties of the classified surfaces. Specifically, **we**

**now adopt the terms "flat terrain" and "rugged terrain" to denote relatively smooth and rough terrain, respectively. Based on this framework, we further subdivided the surface into multiple relief classes, as detailed in the revised manuscript.** This revised terminology is more commonly used in ecological and climate-related research, where terrain relief rather than strict landform taxonomy is emphasized. Furthermore, we have revised our classification system (as detailed in the following response) to improve the rationality and clarity of the classification results. **In addition, we have added descriptions of K1, K2, and K3 in the manuscript to explain the underlying reasons for their differences.** We also agree with your observation that K3 appears to produce reasonable results for the Alps and Himalayas. As shown in Figure 7, our classification exhibits a distribution pattern broadly consistent with K3, while offering improved accuracy in delineating mountainous boundaries.

Following these adjustments, we also updated the released classification dataset, and the associated data repository has been updated accordingly (https://doi.org/10.5281/zenodo.15641257).

[1] Viviroli, D., Kummu, M., Meybeck, M., Kallio, M., & Wada, Y. (2020). Increasing dependence of lowland populations on mountain water resources. Nature Sustainability, 3(11), 917-928.
[2] Meybeck, M., Green, P., & Vörösmarty, C. (2001). A new typology for mountains and other relief classes. Mountain research and development, 21(1), 34-45

The authors must be congratulated on solving the various technical problems in processing a huge combined data set and defining classes that require minimal human editing. The most controversial aspect is the emphasis in level 3 on elevation. This is consistent with the declared aim to be relevant to ecology and human geography. It does mean that the units are bioclimatic as well as geomorphological. A plain is a plain and a drumlin a drumlin whether at 50 m or 4000 m: thus the elevation banding does not give information on landforms. See also Fig. 6e1 comment below.

Because of this, the results may be of more interest to ecologists than to geomorphologists. There should at least be some recognitions of the limitations of using altitude bands within a landform classification.

The Iwahashi classifications produce more complicated patterns, but by using slope, dissection, sinks and terraces they do seem to relate more closely to landform characteristics. From your three levels, I prefer the level 2 to the level 3 result. Alternatively an altitude map and a relief map can be juxtaposed or superimposed.
**Response:**
Thank you for your valuable suggestion. After carefully reviewing your comments and reanalyzing our results, **we agree that incorporating altitude as a classification criterion may introduce certain limitations.** As noted in our previous response, the primary focus of this study is to characterize the vertical variation and relief intensity across different land surfaces. In response to your comments, as well as feedback from the first reviewer and insights from related literature, **we have revised the classification system accordingly.** First, **based**

**on AS, we divided the land surface into two major types: flat terrain and rugged terrain (corresponding to the mountain and lowland categories described in [1]).** Meanwhile, as you noted, introducing elevation-based classification within rugged areas—especially mountainous regions—can result in ambiguous or difficult-to-interpret units in the context of geomorphological or terrain-based research. Therefore, **in the revised manuscript, we have merged the level 2 and 3 within rugged areas.** The updated classification in these regions now emphasizes relative relief, aligning with the suggestion from you. **In flat regions, we retained elevation-based classification, as recommended by previous studies [2,3], to further differentiate plains.** As a result, the revised classification framework now includes two hierarchical levels, capturing the relief characteristics of the land surface in a more interpretable and practical manner. To further enhance clarity and practical value, we have added illustrations in the Data Usage Note section to demonstrate the applicability of our results. We expanded our discussion of how the derived relief classes can support ecological studies—particularly those focused on mountain-lowland runoff balance and water resource allocation.

[1] Viviroli, D., Kummu, M., Meybeck, M., Kallio, M., & Wada, Y. (2020). Increasing dependence of lowland populations on mountain water resources. Nature Sustainability, 3(11), 917-928.
[2] Meybeck, M., Green, P., & Vörösmarty, C. (2001). A new typology for mountains and other relief classes. Mountain research and development, 21(1), 34-45
[3] Zhou, C. H., Cheng, W. M., Qian, J. K., Li, B. Y., & Zhang, B. P. (2009). Research on the classification system of digital land geomorphology of 1: 1000000 in China. Journal of Geo-Information Science, 11(6), 707-724.

DETAILS:
Line 1 As 'models' is plural, perhaps 'multi-source' is unnecessary.
**Response:** We have removed 'multi-source'.

20 'classified landform data' [arguably the DEMs are landform data, so 'lacking' reads strangely.]
**Response:** We have changed it to 'classified relief and landform data'. (Line 22)

27 '26 classes' is mentioned here, but never below. In fact Figs. 4, 5 and 6 and Table A1 show 23 classes, as stated in line 242.
**Response:** Thank you for your suggestion. Although the classification system allows for 26 potential classes, three of them are not observed in practice. Therefore, only 23 classes were presented in the figures and tables. In this revision, we have updated the classification system and eliminated this inconsistency accordingly.

29 Not 'novel'; just expressed here more precisely than before. (We all knew that the Chinese P.R. incorporated very varied landforms).
**Response:** We have changed it to 'finer and more precise spatial disparities in landform patterns than before'. (Line 31)

30 'Peru and China' ? (cf. Fig. 9c)
**Response:** We have changed it to 'Peru and China'. (Line 32)

43 Delete 'the'
**Response:** We have removed 'the' in this sentence.

64 'most previous'
**Response:** We have changed it according to your comment. (Line 31)

71 'increased the complexity of' … 'that poses'.
**Response:** We have changed this sentence according to your comment. (Line 67)

106 Insert 'based on slope'.
**Response:** In the revised manuscript, this sentence has been removed due to the modifications made to the classification scheme. To address your concerns, we have added new statements and incorporated the corresponding descriptions as suggested: "In this study, we refer to the two primary surface types as flat terrain and rugged terrain, based on their differences in slope characteristics." (Lines 110-111)

110 NO: Fig. 4 contradicts this. Plains are subdivided only in level 3. Therefore on –
**Response:** We revised this sentence to "At L2, the flat terrain retains elevation-based characteristics and is further divided into low-altitude, middle-altitude, high-altitude, and very high-altitude flat terrain." (Lines 116-117)

112 'plains, hills and mountains…'
**Response:** This sentence has been removed in the revised manuscript.

121 Omission: REMA must be mentioned / defined.
**Response:** Reference Elevation Model of Antarctica (REMA) and its citation [1] have been added in this paragraph. (Line 126)
[1] Howat, Ian, et al., 2022, "The Reference Elevation Model of Antarctica – Strips, Version 4.1", https://doi.org/10.7910/DVN/X7NDNY, Harvard Dataverse, V1.

144-149 This could be better expressed. 'Transitions have plain cores…' is confusing: transitions are AROUND cores; transitions contain small areas of steeper slopes… The sentences 'Misclassifications…' and 'Meanwhile …' are too vague to be meaningful.
**Response:** We have changed this sentence to "Transitions occur around cores and contain areas with higher slope than typical flat terrain" (Lines 156-157). Meanwhile, the following two sentences have been changed to "In this context, misclassification tends to occur in transitional zones, which exhibit mixed topographic features that do not fully align with either flat or rugged terrain characteristics." (Lines 161-162) and "The boundary is defined as the spatial margin of flat terrain where topographic properties and classification labels shift gradually toward those associated with rugged terrain." (Lines 159-161)

149 Delete 'the'

**Response:** 'the' has been removed in the revised manuscript.

161-166 These very general sentences surely should come earlier?

**Response:** As you suggested, these sentences are indeed better placed earlier in the section. Therefore, we have reorganized Section 2.2.2 accordingly. Specifically, the original content from lines 161–166 has been merged into the first paragraph of this section to clarify the overall conceptual design of the ontology-based method. In the second paragraph, we focus on the specific implementation of this approach using quantitative terrain factors.

168 'very low relief' – but defined by slope, above.

**Response:** We originally used the term relief to characterize surfaces with relatively flat and low-undulating morphological features. In the revised manuscript, we have added 'slope' in this sentence to clarify our idea. (Line 156)

169 'containing' But a better expression of lines 168-169 is 'with elements steeper and others flatter than the threshold slope.'

**Response:** Thank you for your suggestion. As mentioned earlier, we have reorganized the content of Section 2.2.2. In the revised version, the original sentence has been merged into another sentence "Transitions occur around cores and contain areas with higher slope than typical flat terrain, i.e. areas that in part satisfy their classification as flat terrain but also exhibit sloping characteristics not typical of flat terrain. In a general geographic context, these areas should also be classified as flat terrain." (Lines 156-158)

169-171 Another two vague, uninformative sentences. What proportion of the transition area can have non-plain slopes, before it is excluded from 'plain'? 172-178 does not clarify this. The boundary is eventually drawn at the T2 threshold but what are the consequences of that for what is eventually labelled 'plain'?

**Response:** In the revised manuscript, we have reorganized Section 2.2.2 to improve clarity. First, we added a statement in the first paragraph to better explain the relationship between transition areas and plain cores: "In a general geographic context, these areas should also be classified as flat terrain. However, current methods that emphasize local topographic characteristics often fail to identify them correctly" (Lines 158-199). Second, after determining the T2 threshold, regions with AS values lower than T2 are defined as transition areas. These are subsequently merged with the plain cores to form a complete plain unit. We have added further explanation of this process in the revised version to enhance understanding: "Areas where the AS value is less than $T_{AS}$ are merged with the cores to form the complete flat terrain, while the remaining areas are classified as rugged terrain." (Lines 192-193)

181 'typical area extracted'

**Response:** We have revised this sentence to "the core is as extracted in the previous step". (Line 181)

192 what are the units?

**Response:** The AS index is a dimensionless quantity, therefore there is no unit.

It seems T1=Tss and T2=Tas – so why are both versions necessary?
**Response:** In the revised manuscript, we have renamed $T_1$ and $T_2$ to $T_{SS}$ and $T_{AS}$, respectively, to maintain consistency in terminology throughout the text.

194 What was the area of the largest island that required human intervention? Is this the only part of the process that required human intervention, or does variation of the slope threshold between 1.5 º and 3º also require such?
**Response:** In this study, based on our preliminary experiments, we identified that islands with an area smaller than 36 km² require human intervention to ensure accurate classification. Aside from the desert regions already discussed in the manuscript, these small islands are the only areas where manual intervention was necessary. Additionally, we carefully re-examined our experimental procedures and confirmed that the current classification results were all generated using a slope threshold of 1.5 degrees. We apologize for the earlier oversight in describing this aspect. In the revised manuscript, we have added explanations addressing both of these issues to clarify our methodology.

198-199 Only one 'novel', please… delete first.
**Response:** We have deleted it.

215 colon rather than comma after 'limitations'.
**Response:** We have changed it as your suggestion.

217 delete second 'method'
**Response:** We have deleted it.

217 'which does not'
**Response:** We have changed it as your suggestion. (Line 217)

218 'relief relative …'
**Response:** We have changed it as your suggestion. (Line 218)

222 'for objectives' … 'mountain climate' .
**Response:** We have revised it as your suggestion. (Line 222)

229 '3c), we' … 'features,'
**Response:** We have revised it as your suggestion. (Line 229)

233 'Specifically, we construct …'
**Response:** We have revised it as your suggestion. (Line 233)

240 Not "For the plains," but 'For both plains and mountains,'
**Response:** Thank you for your valuable suggestion. In the revised manuscript, we have

adjusted the classification system. Currently, only the flat terrain (formerly referred to as "plain") is classified based on elevation. Accordingly, we have revised the original sentence to specify "for flat terrain" to improve accuracy and clarity.

241 I would describe these as 'classes', rather than landforms.
**Response:** We have revised it as your suggestion. (Line 241)

241 "Mountains are classified as hill, low-relief, middle-relief, high-relief, and highest-relief mountains, based on threshold SRI values of 200m, 1000m, 3500m and 5000m." belongs in the previous section – line 237?
**Response:** In the revised manuscript, we have restructured the content accordingly. Section 2.2.3 (originally around line 237) is now dedicated to quantifying surface relief in the rugged terrain and does not involve specific Level-2 classification thresholds. All threshold values and classification criteria are now presented in Section 2.2.4 to improve clarity and coherence

255-269: this does not seem to cope with the altitude banding problem identified below in Fig. 6e1.
**Response:** As you pointed out, the altitude banding problem may not be fully resolved at this stage. In desert regions, our primary concern is the limited applicability of the previous method. The optimization steps described in this section are mainly designed to extract accurate feature lines to support the calculation of the surface relief index. We have provided a more detailed explanation in our response to your comment regarding Fig. 6e1 below.

Fig.4: What is the projection? Unfortunately it is not equal-area …
**Response:** We have changed it to an equal-area projection 'Equal Earth' (EPSG: 8857).

283 Delete 'the'
**Response:** We have deleted this word.

286 'underscores'
**Response:** We have revised it as your comment. (Line 285)

Fig.5 (see general comment.) Also, why is one side of each dune grey, the other yellow? This cannot be altitude, it mimics the illumination (hill shading) effect seen in the remote sensing image. How can relief vary so much between one side of a dune and the other?
**Response:** The grey and yellow tones observed in Figure 5 are not caused by hill shading effects, but rather reflect the actual color differences of the sand surface captured in the remote sensing imagery. These two color variations do not indicate the two sides of a dune, but instead represent distinct geomorphic units—dunes and interdune areas. As illustrated in the figure below, which corresponds to an adjacent area of Figure 5c, the imagery has not been mosaicked, and thus lighting conditions should be consistent. By comparing with the right-hand side of the image, we can observe that shadows are expressed as dark or black areas, not grey or yellow. Therefore, the "various relief between one side of a dune and the other" mentioned in the comment more accurately reflects the relief between dune crests and interdune flats.

[Figure]

292 Column heading with 'dataset (90 m)' should be 'Iwahashi & Yamasaki' … not et al.
**Response:** We have revised it as your comment. (Line 291)

Fig. 6e1 Why do dunes have a dark brown core, surrounded by a pale brown shade? This distinction seems to arise from an altitude contour intersecting at mid-dune. If so, that is pretty meaningless in terms of classifying landforms: each dune is a single landform. (Presumably the pale green represents inter-dune corridors.)
**Response:** As you pointed out, dunes should be treated as a single and continuous landform. In our Level-1 classification, we aimed to preserve this integrity by distinguishing between dune and interdune areas. We agree that introducing elevation bands may fragment the dune structure. Therefore, in the revised manuscript, we merged elevation and relief information to reduce the adverse effects of altitude-based segmentation. After this revision, we observed that some fragmentation issues still persisted in certain regions. In the updated classification, the segmentation is primarily based on the SRI (Surface Roughness Index), and the classified results could be referred to dune ridges, slopes, and interdune zones. In the revised manuscript, we have added a description of this limitation, acknowledging that the proposed method may, in some cases, generate landform units that do not fully preserve geomorphological completeness.

313 What is the difference in construction of K1 and K2, if both are based on 1000 m resolution DEMs? The coarse resolutions (even the 250 m K3) make comparisons with GBLU's 30 m dubious.
**Response:** K1 was developed for mapping global mountain forests, where mountainous areas are identified based on whether the combined values of elevation, slope, and ruggedness (or relative relief) exceed predefined thresholds. In contrast, K2 was designed to facilitate mountain biodiversity comparisons and adopts a similar approach, but uses ruggedness as the sole classification criterion.

As you rightly noted, there is a resolution difference between K3 and our results. However, the selected dataset K3 currently represents the highest-resolution global dataset available for

mountain classification, and thus serves as the most suitable reference for comparison. To further evaluate the reliability of our dataset, we also compared it with other higher-resolution landform and terrain classification results at different scales. We think that the combination of these comparisons provides a comprehensive validation of our classification outcomes.

300 'Meanwhile valley-like objects'
**Response:** We have revised it as your comment. (Line 299)

337 Some plains (alluvial) are really flat. They must be much less than 70% of land area. Distinguishing these might be more useful than using a purely altitude-based division of plains.
**Response:** As you pointed out, distinguishing plains—especially alluvial plains—is essential. Our team has been exploring methods to extract such features, including recent work (e.g., Sun et al. 2025). However, considerable challenges remain when addressing global-scale classifications. We hope that in future work, by incorporating additional data layers into the current dataset, we can further refine the classification to distinguish genetically meaningful units such as fluvial/alluvial plains.

Sun, W., Chen, Y., Zhou, X., Yang, X., Ma, J., Li, S., & Tang, G. (2025). Understanding the hydrological valley landscape: A multi-scenario adaptive framework for delineating valley floors. Catena, 256, 109111.

351 This heading is uninformative. The section deals with 'Geographic relationships: climate and land use'.
**Response:** According to your comment and the new experiment, we have changed the heading of this section to 'Geographic relationships: runoff, climate and land use'. (Line 343)

352 'relationships'
**Response:** We have revised it as your comment. (Line 350)

364 This is mainly due to ice sheets ! They should be considered separately.
**Response:** Thank you for your valuable comment. We have included additional explanations in the manuscript to highlight the distinct characteristics of this region and to clarify the special influence of ice sheets on surface elevation and relief patterns. (Lines 372-375)

391-392 Only one 'novel', please… delete second?
**Response:** We have revised it as your comment.

Table A3 Two names are uninformative or inaccurate:
'Great Socialist People' is 'Libya'. The full name is 'State of Libya'.
Checking where ' Democratic People' might be, I find that DZA is 'Algeria'.
**Response:** We have checked these names and revised these mistakes.

---

## Author Response (AR3)

Dear Authors,

Thank you for considering the comments from the previous round of reviews thoroughly, and for adapting your manuscript accordingly.

I am now satisfied that the manuscript is suitable for publication in ESSD.

However, for transparency and reproducibility, please create a GitHub release (or tag) corresponding to the exact version of the code used for this paper. Ideally, archive this version with Zenodo to obtain a DOI, and cite it in the data descriptor. This ensures future readers can access the code in the same state as during the analysis.

Kind regards,

James

**Response:** Thanks to you and the reviewers for your valuable suggestions and efforts during the review process! In the revised version, we have added and updated the code in GitHub (https://github.com/nnu-dta/Global-Relief-Classes) corresponding to the exact version. Some of the steps were developed based on ArcGIS Pro, which may involve commercial licensing issues. We are developing and updating them based on open-source code. Meanwhile, we have added the GitHub repository link to the manuscript (Section: Dataset access) and the Zenodo link.